# CIDD: Collaborative Intelligence for Structure-Based Drug Design Empowered by LLMs

**Bowen Gao[1,2]\***, **Yanwen Huang[3]\***, **Yiqiao Liu[3]**, **Wenxuan Xie[4]**, **Bowei He[5]**, **Haichuan Tan[1]**,
**Wei-Ying Ma[1]**, **Ya-Qin Zhang[1]**, **Yanyan Lan[1,6,7]†**

[1]Institute for AI Industry Research (AIR), Tsinghua University, Beijing, China
[2]Department of Computer Science and Technology, Tsinghua University, Beijing, China
[3]Department of Pharmaceutical Science, Peking University, Beijing, China
[4]School of Future Technology, South China University of Technology, Guangzhou, China
[5]Department of Computer Science, City University of Hong Kong, Hong Kong SAR, China
[6]Beijing Frontier Research Center for Biological Structure, Tsinghua University, Beijing, China
[7]Beijing Academy of Artificial Intelligence (BAAI), Beijing, China

## Abstract

Structure-guided molecular generation is pivotal in early-stage drug discovery, enabling the design of compounds tailored to specific protein targets. However, despite recent advances in 3D generative modeling, particularly in improving docking scores, these methods often produce uncommon and intrinsically unreasonable molecular structures that deviate from drug-like chemical space. To quantify this issue, we propose a novel metric, the Molecule Reasonable Ratio (MRR), which measures structural rationality and reveals a critical gap between existing models and real-world approved drugs. To address this, we introduce the Collaborative Intelligence Drug Design (CIDD) framework, the first approach to unify the 3D interaction modeling capabilities of generative models with the general knowledge and reasoning power of large language models (LLMs). By leveraging LLM-based Chain-of-Thought reasoning, CIDD generates molecules that are not only compatible with protein pockets but also exhibit favorable drug-likeness, structural rationality, and synthetic accessibility. On the CrossDocked2020 benchmark, CIDD consistently improves drug-likeness metrics, including QED, SA, and MRR, across different base generative models, while maintaining competitive binding affinity. Notably, it raises the combined success rate (balancing drug-likeness and binding) from 15.72% to 34.59%, more than doubling previous results. These findings demonstrate the value of integrating knowledge reasoning with geometric generation to advance AI-driven drug design.[3]

## 1   Introduction

Structure-based drug design (SBDD) enables the direct generation of compounds tailored to protein binding sites, making it a powerful tool in drug discovery. Recent 3D generative models—including autoregressive approaches like AR [24] and Pocket2Mol [25], and diffusion-based methods such as TargetDiff [12], DecompDiff [13], and MolCRAFT [26]—have advanced rational molecule design. However, expert evaluation often reveals chemically unreasonable features, such as overly fused polycyclic systems and partially unsaturated rings (Figure 1a). While these distortions may improve

---

\*Equal contribution
†Correspondence to `lanyanyan@air.tsinghua.edu.cn`
[3]Code is available at https://github.com/bowen-gao/CIDD/.

docking scores via geometric complementarity, they often compromise intramolecular stability and result in poor pharmacokinetic properties.

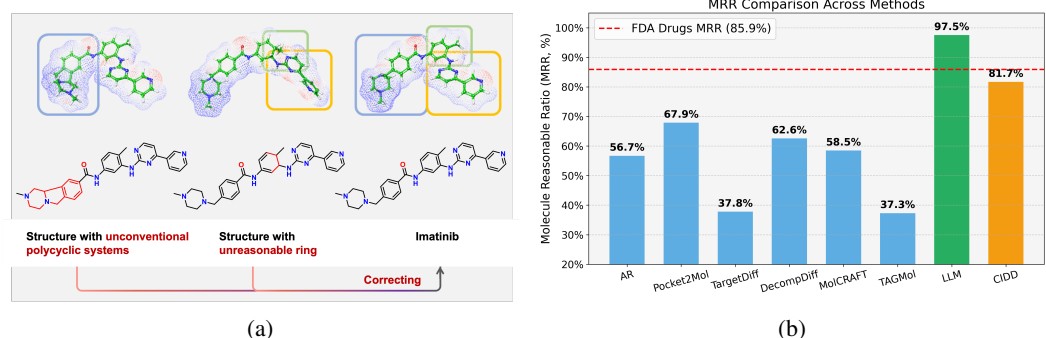

Figure 1: (a) **Common errors in 3D-SBDD outputs.** Minor structural changes can cause large deviations in 3D conformation, highlighting the challenge of correcting chemically uncommon structures without disrupting valid 3D shapes. (b) **MRR comparison.** While FDA drugs reach 85.9% MRR, existing 3D models lag behind. LLM achieves 97.5%, and CIDD closes the gap with 81.7%.

While current models effectively generate molecules that fit target pockets, they often overlook the physicochemical and pharmacological features common to clinically approved drugs. These limitations stem from a fundamental mismatch: the narrow focus on binding-site compatibility often comes at the expense of structural reasonability and drug-likeness, which depend solely on the chemical structure itself and often conflict with geometric fit. To quantify the gap, we propose the **Molecular Reasonability Ratio (MRR)**, which measures the proportion of generated molecules that are structurally valid and chemically reasonable relative to drug-like standards. As shown in Figure 1b, based on our experiments, existing 3D generative models perform poorly under this metric (e.g., TargetDiff: 37.8%, MolCRAFT: 58.5%), falling well below the 85.9% observed in FDA-approved drugs. Some recent methods, such as TAGMol, build on TargetDiff by incorporating gradient-based guidance to jointly optimize for drug-likeness properties like QED [3]. While this leads to improvements in QED scores, it does not result in a general enhancement of drug potential, as evidenced by TAGMol's similarly low MRR. In contrast, LLMs achieve high MRRs on SBDD tasks (e.g., GPT-4: 97.5%) thanks to their general chemical and pharmaceutical knowledge, but they struggle to generate molecules with high binding affinity due to limited spatial reasoning and the inherent trade-off between potency and drug-likeness.

To bridge this gap, we propose **CIDD**—a **Collaborative Intelligence for Drug Design** framework that combines the interaction modeling capabilities of 3D generative models with the domain knowledge and instruction following strengths of LLMs. Rather than directly producing final drug candidates, 3D models generate interaction-focused molecular proposals that capture key binding features. These serve as structured input for an LLM-driven design process that translates spatial intent into chemically viable drug-like molecules. This division of roles allows CIDD to generate compounds that are both interaction-competent and structurally plausible. CIDD decomposes the generation task into a series of specialized modules, each powered by the LLM. The *Interaction Analysis Module* identifies fragment-level interactions; the *Design Module* proposes chemically informed modifications; the *Reflection Module* provides analysis; and the *Selection Module* ranks candidates by interaction quality and chemical coherence. CIDD follows a structured **Chain-of-Thought (CoT)** prompting strategy guided by domain knowledge guided prompts, enabling interpretable, stepwise molecular design. **This modular, CoT-guided architecture mirrors real-world medicinal chemistry workflows and enables the generation of pharmaceutically relevant, structurally sound compounds.**

Evaluated on the CrossDocked2020 dataset [9], CIDD significantly outperforms state-of-the-art baselines, boosting the overall success ratio from **15.72% to 34.59%**. It consistently improves key drug-likeness metrics—including *Quantitative Estimation of Drug-likeness* (QED) [3], synthetic accessibility (SA), MRR, and the proportion of molecules meeting QikProp [30] thresholds—across diverse baseline models. **Our key contributions are:**

(1) **Identifying a core limitation in current SBDD models:** We introduce the **MRR** to assess intrinsic structural rationality, revealing that pocket-based 3D generative models often produce

*intrinsically irrational* and rare molecules that deviate from drug-like chemical space despite favorable docking scores.

(2) **A unified framework for rational molecule generation:** We propose **CIDD**, the first framework to combine the spatial modeling of 3D generative models with the chemical reasoning capabilities of LLMs. By leveraging their complementary strengths, CIDD uses Chain-of-Thought guidance to overcome the common trade-off between binding affinity and drug-likeness.

(3) **Validated performance and broader implications:** CIDD achieves state-of-the-art results on CrossDocked2020, significantly improving drug-likeness metrics (QED, SA, MRR) while preserving binding affinity. This demonstrates its effectiveness and highlights the advantage of combining LLMs with geometric modeling for AI-driven drug design.

## 2 Preliminaries

### 2.1 Structure-Based Drug Design

The goal of SBDD is to generate a molecule $x$ that binds to a given protein pocket $P$. Recent work has explored a variety of deep generative approaches. LiGAN [27] uses a conditional variational autoencoder (CVAE) to generate 3D point clouds of ligands conditioned on pocket geometry. Autoregressive models such as AR [24] and Pocket2Mol [25] sequentially construct molecules in 3D space. Diffusion-based methods like TargetDiff [12], IPDiff [15], and DecompDiff [13] iteratively refine 3D structures or molecular graphs, while TAGMol [7] further applies gradient-based optimization during generation. Other approaches include the fragment-based DrugGPS [37] and MolCRAFT [26], a Bayesian flow model that learns the distribution of drug-like molecules conditioned on pocket structures. All these methods generate intermediate 3D point clouds or graphs, which are subsequently mapped into chemically valid molecules as the final output.

### 2.2 Large Language Models

LLMs are advanced neural networks trained on vast corpora of textual data, enabling them to understand, generate, and reason with human-like language. Notable examples include GPT-4 [2], LLaMA [33], ChatGLM [11], and DeepSeek [21], which have demonstrated impressive capabilities in tasks such as natural language understanding, code generation, mathematical problem solving, and logical reasoning. Their versatility and ability to capture complex patterns have made them increasingly relevant in domains beyond traditional natural language processing, including drug discovery [4]. Recent work has explored the use of LLMs to generate or modify molecules with desired properties. For example, ChatDrug [22] employs conversational LLMs for generating and editing molecules with desired properties. However, the application of LLMs to *pocket-based* molecular design remains largely unexplored. This is primarily due to the challenge of capturing essential three-dimensional structural information, which is critical for modeling protein-ligand interactions. Unlike textual or sequence-based data, protein binding pockets and small molecules interact through spatially complex, non-linear, and chemically rich environments that cannot be directly encoded in natural language.

## 3 Methods

### 3.1 Evaluating the Gap Between Generated Molecules and Real Drugs

Drug-likeness is an inherently multidimensional attribute that dictates the probability $p(\text{drug})$ that a molecule will reach its biological target—an assessment that precedes, and is distinct from, estimating binding affinity. Widely used machine-learning surrogates, most notably QED [3] and Lipinski's Rule of Five [20], are inadequate for two principal reasons: (i) Legacy bias: These metrics were calibrated on historical chemical space and fail to recognize the unconventional scaffolds produced by modern generative models. Consequently, structurally implausible or pharmacologically irrelevant molecules can still achieve high "drug-like" scores. (ii) Oversimplification: QED relies on only seven physicochemical descriptors, and its average value for DrugBank [17] molecules is below 0.5, illustrating its limited discriminative power.

To close this gap, we introduce a broader evaluation framework comprising two complementary tiers. Structural reasonability screens for fundamental medicinal-chemistry principles such as aromaticity, ring-system stability, and hybridization consistency, while physicochemical and pharmacokinetic suitability evaluates absorption, distribution, metabolism, excretion, and toxicity (ADME/T) properties using established descriptors. Together, these tiers ensure that generated molecules are both chemically plausible and therapeutically relevant, providing a more reliable benchmark for modern AI-driven molecular design.

**Molecular Reasonability Ratio.**     To address the lack of metrics capable of identifying the structural gap between generated molecules and rationally designed drugs, we introduce MRR as a targeted solution. MRR is a rule-based, empirically motivated metric designed to flag structures that are uncommon in medicinal chemistry practice. Its design is informed by expert analyses of SBDD outputs, which consistently reveal a predominant failure mode: the improper use of ring conjugation—particularly aromaticity—that often deviates from patterns observed in clinically validated drugs.

MRR operates by analyzing the hybridization states of atoms within a molecule. It sequentially excludes fully aromatic or fully saturated rings and focuses on the remaining atoms within each ring system. After removing peripheral substituents, if the hybridization states of the residual ring atoms are neither all $sp^2$ nor all $sp^3$, the structure is classified as chemically unreasonable. For example, structures such as cyclohexene and cyclohexa-1,3-diene are flagged, as they likely result from the model's failure to correctly generate either fully aromatic or fully saturated systems. Although not intended as an exhaustive classifier, MRR provides an interpretable, medicinal-chemistry-guided screen that complements basic valence checks and ADME filters, thereby reducing the incidence of structurally implausible candidates generated by models. The full algorithm for MRR is shown in Appendix D.

**QikProp Multiple Property Requirements.**     To further evaluate the physicochemical and pharmacokinetic properties of the generated molecules, we employ QikProp [30], a tool recognized for its robust performance in predicting molecular drug-likeness properties [16]. The assessed properties include aqueous solubility, lipophilicity, polar surface area (PSA), the number of metabolizable sites, and oral absorption. Detailed requirements for each property are provided in Appendix E.

A molecule is considered to have passed the evaluation if it satisfies all $N$ predefined property requirements: $P_1, P_2, \ldots, P_N$. If any of the properties fall outside the acceptable range, the molecule is classified as failing.

$$\text{QikProp} = \begin{cases} 1 & \text{if } P_1 \wedge P_2 \wedge \cdots \wedge P_N \text{ are satisfied,} \\ 0 & \text{otherwise.} \end{cases}$$

### 3.2   Bridging the Gap with CIDD framework

We propose the **Collaborative Intelligence Drug Design (CIDD)** framework (Figure 2), a modular system for target-specific molecule generation that combines 3D interaction modeling with LLM-enhanced molecular design. It consists of two components: the **Structure-Based Interaction Generator (SBIG)** and the **LLM-Enhanced Drug Designer (LEDD)**.

**SBIG** uses 3D generative models to produce *interaction-oriented molecular structures*—intermediates that fit the protein pocket but might be chemically incomplete.

**LEDD** then refines these raw proposals into drug-like molecules using LLMs. The whole process is formalized as:
$$x_0 = \texttt{SBIG}(\textit{Target}), \quad x = \texttt{LEDD}(x_0, \textit{Target})$$

Here, $x_0$ encodes the intended spatial interactions, and LEDD completes it into a chemically valid, synthetically accessible compound $x$. A key limitation of prior approaches is the absence of ground-truth mappings from interaction scaffolds to drug-like molecules, making supervised training infeasible. Traditional generative models—based on direct optimization or limited heuristics—struggle to bridge this gap. In contrast, LEDD leverages the implicit drug-relevant knowledge embedded in large-scale pretrained LLMs. With strong **instruction-following, reasoning, and coordination abilities**,

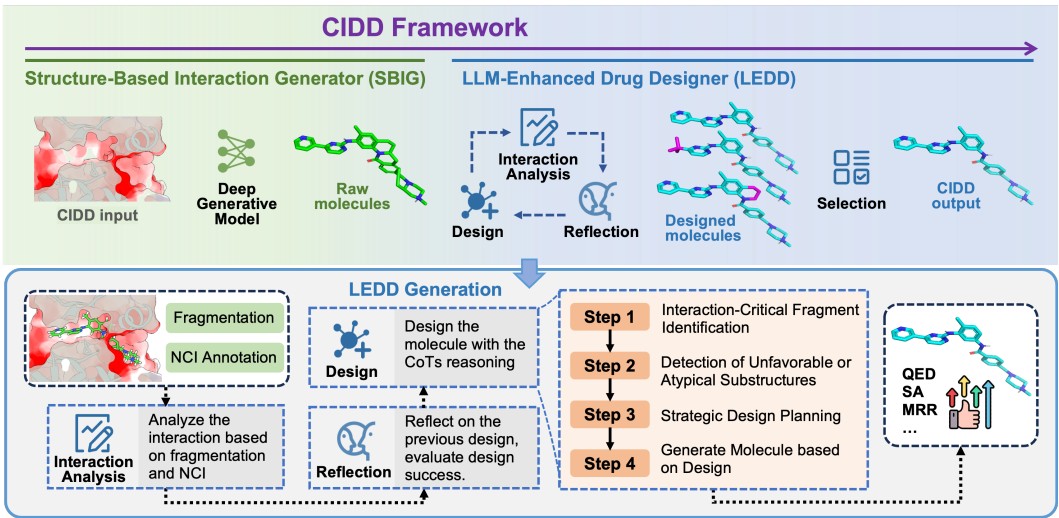

Figure 2: **Overview of the CIDD framework.** Top: end-to-end pipeline integrating SBIG and LEDD. Bottom: detailed view of LEDD's LLM-based design process, which designs and generates molecules through interaction analysis, reflection, and Chain-of-Thought-driven reasoning.

the LLM enables design decisions that generate molecules without incurring the typical trade-offs between potency and drug-likeness in drug design.

The CIDD framework is inspired by the real-world workflow of medicinal chemists, who begin by identifying key interaction patterns between a ligand and its target, and then construct molecules that retain these interactions while satisfying broader drug development constraints. CIDD formalizes this reasoning process: SBIG captures spatial interaction intent, while LEDD—powered by LLMs—assumes the role of a virtual chemist to transform raw scaffolds into viable drug-like molecules. By leveraging the LLM's instruction-following and general knowledge, CIDD effectively substitutes human expertise in the molecule design process.

### 3.2.1 Pipeline Overview

CIDD decomposes the complex task of drug-oriented molecule generation into a structured pipeline of modular reasoning steps. It begins with SBIG, which generates interaction-focused intermediate structures conditioned on the target pocket. These intermediates are passed to the **Interaction Analysis Module**, which extracts fragment-level binding profiles using docking, fragmentation, and rule-based NCI detection. The **Design Module** then interprets these profiles to generate chemically plausible molecules that maintain critical interaction motifs. Each design is re-evaluated, and its updated interaction profile is processed by the **Reflection Module**, which compares successive design states and produces feedback for new designs. After generating $N$ new candidates, the **Selection Module** ranks all candidates based on their interaction quality and chemical viability, selecting the final molecule for downstream use. This modular pipeline mirrors expert workflows and enables interpretable, multi-stage control over both structural interaction and chemical relevance.

### 3.2.2 Interaction Analysis Module

This module evaluates the molecular interaction between a candidate molecule $x_i$ and a target protein pocket $P$. The process begins by re-docking $x_i$ into the binding site of $P$ to generate a protein–ligand complex conformation. The molecule $x_i$ is then decomposed into chemically meaningful fragments via the BRICS algorithm [6], enabling fragment-level attribution of interaction features.

Next, non-covalent interactions (NCIs), such as hydrogen bonding, $\pi$–$\pi$ stacking, salt bridges, and hydrophobic contacts, are identified from the docked complex based on geometric and chemical features. The resulting interaction map—capturing both atomic contacts and fragment-level organization—is then analyzed by an interaction-specialized large language model, $LLM_I$. Conditioned on spatial interaction cues and chemical context, $LLM_I$ synthesizes this information into a structured

interaction profile $I_i$, which semantically links molecular fragments to their respective roles in target engagement.

Formally, the process can be represented as:

$$\text{LLM}_I(x_i, P) \rightarrow I_i$$

The resulting profile $I_i$ provides fragment-level interpretation of binding contributions and serves as the basis for chemically informed design in subsequent modules.

### 3.2.3 Design Module

Given a raw molecule $x_0$, its interaction report $I_0$, and prior feedback from the reflection module $R = \{R_1, R_2, \ldots, R_{i-1}\}$, the Design Module leverages an LLM with a Design role ($LLM_D$) to produce a refinement plan $D_i$ and an updated molecule $x_i$.

$$LLM_D(x_0, I_0, R) \rightarrow (D_i, x_i)$$

Here, $D_i$ is a structured and interpretable set of design decisions grounded in medicinal chemistry, and $x_i$ is the generated compound for further evaluation.

To emulate real-world drug design workflows, we implement a **Chain-of-Thought** prompting strategy that guides the LLM through a domain-informed reasoning pipeline with four sequential stages. **(1) Interaction-Critical Fragment Identification:** the model analyzes $x_0$ and highlights fragments forming key non-covalent interactions with the target, based on $I_0$. **(2) Detection of Unfavorable or Atypical Substructures:** chemically undesirable fragments—such as synthetically inaccessible motifs, strained rings, or poor physicochemical regions—are flagged for replacement. **(3) Strategic Design Planning:** the model proposes modifications that improve chemical viability while preserving interaction and topology, often substituting problematic regions with pharmaceutically preferred alternatives. **(4) Candidate Molecule Generation:** a new structure $x_i$ is generated based on the design plan, ensuring both target interaction and drug-like properties.

By encoding expert priors into the prompt structure, the Design Module reframes molecule generation as a chemically grounded reasoning task. The LLM is able to coordinate multiple design objectives within a unified framework. This enables the CIDD to effectively mitigate the trade-offs between affinity and drug-likeness that commonly arise in drug design. It bridges raw interaction scaffolds to viable candidate molecules in a transparent and interpretable manner.

### 3.2.4 Reflection and Selection Modules

The *Reflection Module* evaluates whether each candidate design $x_i$ achieves design goals by comparing it to the initial raw molecule $x_0$ and their respective interaction profiles:

$$LLM_R(x_0, I_0, D_i, x_i, I_i) \rightarrow R_i$$

The *Selection Module* evaluate all the designed molecules $\{x_1, x_2, \ldots, x_N\}$ and their interaction reports to identify the most promising candidate:

$$LLM_S\left(\{(x_1, I_1), \ldots, (x_N, I_N)\}\right) \rightarrow x_{\text{best}}$$

Here, $x_{\text{best}}$ balances interaction quality and chemical viability. Full prompt examples for each module are in Appendix C.

## 4 Experiments

### 4.1 Experiment Settings

**Dataset.** We follow prior 3D-SBDD settings and use the CrossDocked2020 dataset [9], adopting the same train/test split as TargetDiff [12], resulting in 100 protein pockets for test.

**Metrics.** We evaluate models using standard metrics: Vina docking score [34] for binding affinity, QED [3] for drug-likeness, SA score [8] for synthetic accessibility, and molecular diversity (computed as $1 - $ ECFP4 similarity [29]) for structural variety. As we do not directly generate 3D conformations,

Vina Score/Min is excluded. Beyond conventional metrics, we emphasize the evaluation of **drug potential** via two additional metrics: **MRR (Molecular Rule-based Reasonability)**, which captures domain-informed structural plausibility, and **QikProp pass ratio**, which assesses a wide range of physicochemical and pharmacokinetic properties. Together with QED and SA, these offer a more holistic assessment of molecular viability. We also report the success ratio, defined as the percentage of molecules that satisfy all of the following criteria: Vina $< -8.18$, QED $> 0.25$, and SA $> 0.59$, following [23]. In addition, we introduce molecular reasonability as an extra constraint for defining success. We also evaluate the proportion of molecules that pass all QikProp filters.

**Baseline Models.** We compare CIDD against a diverse set of 3D-SBDD baselines spanning multiple generative paradigms: the VAE-based **LiGAN** [27]; autoregressive models **AR** [24] and **Pocket2Mol** [25]; and diffusion-based approaches including **TargetDiff** [12], **IPDiff** [15], **DecompDiff** [13], and **TAGMol** [7]. **IPDiff** uses interaction-guided sampling, while **DecompDiff** leverages interaction-aware priors; **TAGMol** adds gradient-based optimization for drug-likeness. We also evaluate **DrugGPS** [37], a fragment-based method, and **MolCRAFT** [26], a Bayesian flow network shown to be state-of-the-art in recent benchmarks [19, 10].

**CIDD Settings.** We use MolCRAFT in the SBIG step. All modules in the LEDD step are powered by GPT-4o. The Design Module generates 5 candidates per round, and the Selection Module selects the final molecule. For each protein pocket, we generate 10 molecules. All SBIG models are trained on CrossDocked2020 with their released weights.

## 4.2 General Results

As shown in Table 1, CIDD demonstrates strong overall performance across key drug-likeness metrics—including QED, MRR, SA, and QikProp pass ratio—as well as favorable binding affinity. In comparison, IPDiff, which leverages a trained binding affinity predictor to guide both training and sampling, achieves some improvement in docking scores over TargetDiff. However, it performs worse in terms of MRR, highlighting the potential limitations of focusing solely on binding affinity. Similarly, the gradient-guided method TAGMol aims to enhance multiple properties through predictor guidance during sampling. While TAGMol shows notable improvements in QED, it brings minimal gains and still performs badly in metrics such as MRR and SA, performing comparably to the unguided diffusion model TargetDiff. This suggests that optimization-driven approaches like TAGMol may overfit individual scoring functions rather than learning to generate molecules with generally improved drug-like profiles. In contrast, CIDD delivers **consistent improvements across all major drug-likeness dimensions**, indicating a more robust and comprehensive molecular design capability. These results stem from CIDD's ability to integrate the complementary strengths of distinct modeling

Table 1: **Test Results on CrossDocked2020.** We benchmark several evaluation metrics, including Vina docking score, QED, SA, MRR, success ratio, and QikProp pass ratio. We also report the average molecular weight. Performance ranking per column is color-coded as follows: best , second-best

| Category | Method | Vina ↓ | QED ↑ | SA ↑ | MRR ↑ | Success ↑ | QikProp ↑ | MW |
|---|---|---|---|---|---|---|---|---|
| VAE-based | LiGAN | -6.640 | 0.394 | 0.601 | 59.08% | 2.79% | 17.37% | 286.44 |
| AR-based | AR | -6.737 | 0.507 | 0.635 | 56.67% | 3.28% | 18.66% | 247.50 |
| AR-based | Pocket2Mol | -7.246 | 0.573 | **0.758** | 67.88% | 14.60% | 29.58% | 234.30 |
| Diffusion-based | TargetDiff | -7.452 | 0.474 | 0.579 | 37.81% | 3.04% | 27.63% | 346.24 |
| Diff + Inter-Guide | IPDiff | -7.745 | 0.511 | 0.627 | 29.83% | 5.31% | 25.11% | 328.34 |
| Diff + Inter-Prior | DecompDiff | -8.260 | 0.444 | 0.609 | 62.60% | 15.72% | 29.04% | 424.09 |
| Diff + Multi-Guide | TAGMol | -7.563 | 0.563 | 0.583 | 37.31% | 3.23% | 32.31% | 325.50 |
| Fragment-based | DrugGPS | -7.396 | 0.463 | 0.622 | 54.80% | 7.17% | 25.60% | 329.88 |
| BFN-based | MolCRAFT | -7.783 | 0.503 | 0.685 | 58.47% | 13.72% | 22.37% | 325.63 |
| 3DSBDD + LLM | CIDD | **-8.496** | **0.576** | 0.735 | **81.74%** | **34.59%** | **35.22%** | 336.70 |

paradigms—combining the strong binding affinity modeling capability of 3D generative models with the drug-likeness reasoning and instruction-following power of large language models. Guided by expert-designed prompts and leveraging the LLM's embedded chemical knowledge, CIDD refines molecular structures toward realistic and pharmacologically meaningful candidates. As a result, it achieves a significantly higher success ratio of **34.59%**, compared to **15.72%** for the best

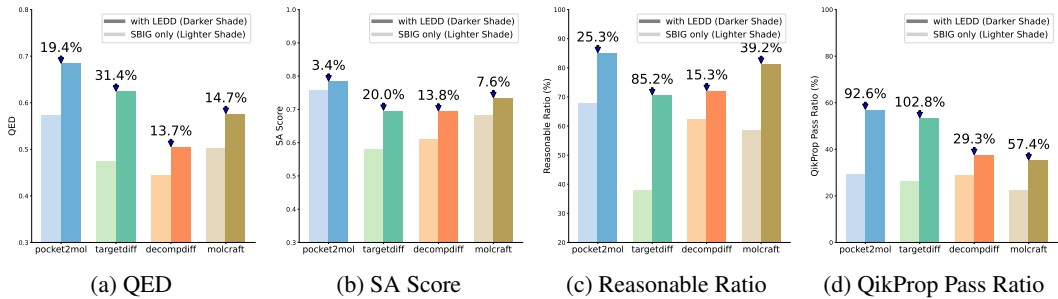

| (a) QED | (b) SA Score | (c) Reasonable Ratio | (d) QikProp Pass Ratio |

Figure 3: Performance comparison across different metrics indicative of drug potential between models using only SBIG outputs and those incorporating LEDD outputs, evaluated across various 3D-SBDD models as the SBIG module.

baseline—demonstrating its unique capacity to generate molecules that are not only strong binders, but also truly drug-like and synthetically feasible. Interestingly, CIDD achieves comparable QED and SA scores, while significantly outperforming Pocket2Mol in MRR, despite generating molecules that are approximately **50% larger** on average in terms of molecular weight (336.70 vs. 234.30). Moreover, it achieves a substantially better docking score. This result provides strong evidence that CIDD is capable of generating **larger molecules that are still drug-like**, suggesting that its strong performance on drug-likeness metrics is not simply a result of hacking these metrics by producing smaller, simpler molecules.

## 4.3 Improvements with Different Models on Multiple Metrics

CIDD is a flexible framework designed to interface smoothly with a broad spectrum of 3D SBDD models, significantly enhancing the quality of generated molecules. As illustrated in Figure 3, CIDD brings substantial and consistent improvements across key drug-likeness metrics—including QED, SA Score, Reasonable Ratio, and QikProp Pass Ratio—achieving gains of 31.4%, 20.0%, 85.2%, and 102.8%, respectively. These improvements are observed across different base models, demonstrating CIDD's strong generalization ability and its capacity to enhance diverse, diverse aspects of drug-likeness simultaneously. In contrast to optimization-based methods that often overfit individual metrics, CIDD drives broad and meaningful improvements that reflect a true advancement in the quality of generated drug candidates.

## 4.4 Ablation and Analysis

Table 2: Ablation studies on LLM variants and pure LLM-based SBDD.

(a) Different LLM Backends in CIDD

| LLM | Vina↓ | MRR↑ | Similarity↑ |
|---|---|---|---|
| - | -7.78 | 58.47% | - |
| GPT-4o-mini | -8.29 | 80.02% | 0.220 |
| GPT-4o | -8.50 | 81.37% | 0.296 |
| DeepSeek-v3 | -8.49 | 76.00% | 0.379 |
| DeepSeek-r1 | -8.57 | 79.17% | 0.182 |

(b) LLM-Only vs. CIDD Comparison

| | Vina↓ | MRR↑ | Success Ratio↑ |
|---|---|---|---|
| LLM-SBDD | -6.244 | 97.45% | 5.95% |
| CIDD-LLM | -7.230 | 90.97% | 17.59% |
| CIDD | -8.496 | 81.74% | 35.22% |

### 4.4.1 Impact of Different LLMs

We evaluate GPT-4o, GPT-4o-mini, DeepSeek-v3 [5], and DeepSeek-r1 [14] using MolCRAFT as the SBIG module (Table 2a). All models improve drug-likeness metrics (MRR, QikProp) and docking scores. DeepSeek-v3 achieves property gains with minimal edits, while GPT-4o-mini struggles with similarity, and DeepSeek-r1 makes broader, less controllable changes. GPT-4o and DeepSeek-v3 best support CIDD's goal of generating similar yet improved molecules. Smaller models like LLAMA-7B fail to follow design instructions. CIDD remains plug-and-play, benefiting from future LLM advances.

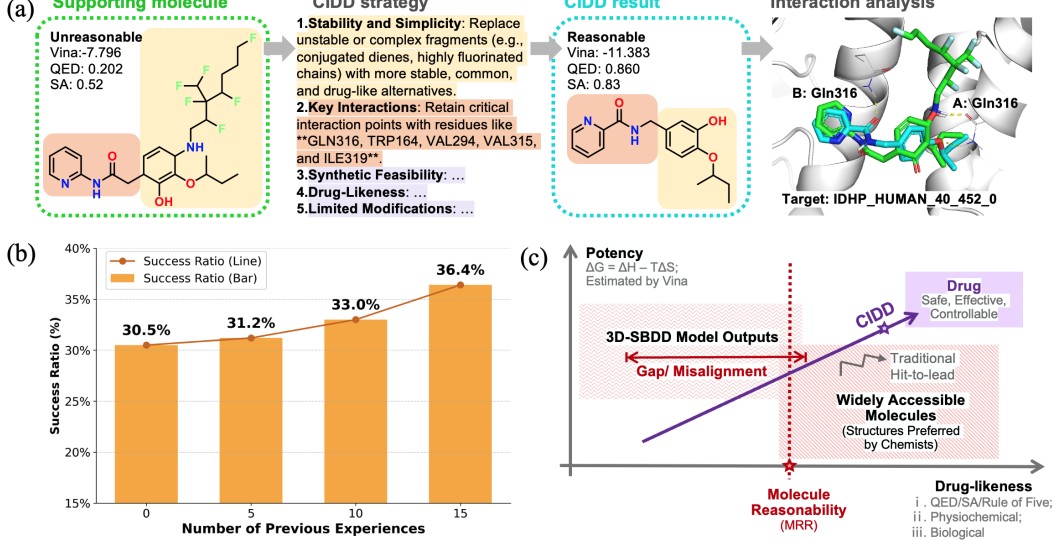

Figure 4: (a) A generation case and corresponding interpretable design strategy produced by CIDD, resulting in a structurally similar yet better compound. (b) CIDD demonstrates the ability to evolve by leveraging previous design experiences as context, improving generation success rates. (c) CIDD integrates the strengths of 3D-SBDD models and LLMs to enable practical drug design with both high potency and drug-likeness.

### 4.4.2 Using Pure LLM for SBDD

LLMs alone struggle with 3D protein pocket interpretation in structure-based drug design (SBDD). To test this, we prompted an LLM with PDB-format pocket data and also evaluated CIDD-LLM, which uses LLM-SBDD within the CIDD framework. As shown in Table 2b, LLM-SBDD generates chemically reasonable molecules but performs poorly on binding affinty, leading to a lower success rate than standard CIDD. This highlights the need for combining 3D models' interaction modeling with LLMs' reasoning.

### 4.5 Advantages and Impact of the CIDD Framework

**Interpretable Molecule Design.** Figure 4a illustrates the CIDD generation process. The LLM-powered modules analyze and refine the raw supporting molecule (green), producing a high-quality final structure (blue). Problematic fragments—such as an unreasonable diene or an uncommon fluorinated chain—are automatically identified and replaced (e.g., with a benzene ring), while side chains are adjusted to preserve key hydrogen bonds with Gln316 on both Chain A and B. These edits improve docking scores and enhance drug-relevant properties. CIDD performs this refinement through localized fragment substitutions, maintaining the core structure while improving overall drug potential. Notably, the process is inherently **interpretable**: each design step is traceable, with explicit rationales highlighting structural strengths and weaknesses. This transforms conventional opaque SBDD into a transparent, expert-assisting workflow. CIDD also enables the **automated creation of molecule pairs** that differ in drug-likeness with minimal structural edits. These pairs effectively capture how small chemical changes influence pharmaceutical viability, offering high-quality, distributionally aligned data for fine-tuning. Compared to random sampling, they provide more meaningful supervision and help mitigate data scarcity in 3D-SBDD (see Appendix H).

**Evolvement Ability.** One key advantage of LLMs is their ability to leverage prior experience, provided as context, to generate insightful outputs. To illustrate this, we conducted a proof-of-concept experiment using a relatively lightweight LLM: GPT-4o-mini. The model was given varying numbers of previous design reports and results (0, 5, 10, and 15) as contextual input and tasked with generating insights to support the design module of CIDD. As shown in Figure 4b, the success rate improved as more prior reports were included. These results demonstrate CIDD's capacity for continual evolution by incorporating accumulated experience—without requiring model retraining. This mirrors the way human experts enhance their performance through repeated exposure and practice.

**Generating Small Molecules with Both High Potency and Drug-Likeness.** Drug potential hinges on two key factors: potency and drug-likeness. While most 3D-SBDD models emphasize target fit, they often produce chemically unreasonable structures. As shown in Figure 4c, our **CIDD** framework bridges this gap by combining geometric modeling with LLM-driven reasoning. The LLM not only corrects unfavorable fragments but also plans coherent molecular edits that balance multiple objectives. By coordinating spatial and chemical constraints within a unified generation process, CIDD effectively overcomes the traditional trade-off between interaction strength and drug-likeness.

## 5 Conclusion

We presented **CIDD**, a collaborative framework that unifies 3D interaction modeling and LLM-driven reasoning for structure-based drug design. CIDD addresses a key limitation of current generative models: the tendency to generate interaction-compatible but chemically unreasonable molecules. Through a modular, interpretable generation process, CIDD achieves state-of-the-art results on the CrossDocked2020 benchmark—substantially improving drug-likeness metrics (QED, SA, MRR, QikProp) while maintaining high binding affinity. By bridging 3D geometric modeling with language-guided design, our approach sets a foundation for future directions in rational, interpretable, and generalizable drug generation. We envision such a collaborative paradigm enabling broader tasks such as target discovery and hit-to-lead optimization in early-stage drug discovery.

## Acknowledgements

This work is supported by Beijing Academy of Artificial Intelligence and Beijing Frontier Research Center for Biological Structure Fundings.

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

## A Discussion on the Pharmaceutical Terminology

In this work, we employ a comprehensive set of evaluation metrics—**Vina**, **QED**, **SA**, our proposed **MRR**, the **Success Ratio**, and **QikProp**—to analyze different facets of molecular generation performance. Following established practices, we use Vina scores to assess the 3D complementarity between generated molecules and their target binding sites. Meanwhile, we evaluate broader molecular properties using standard metrics (QED and SA), the domain-informed MRR, and physicochemical and pharmacokinetic descriptors from QikProp, which are widely used in computer-aided drug design (CADD).

To clarify the key concepts underpinning our evaluation framework, we distinguish among **drug potential**, **drug-likeness**, **molecular reasonability**, and **chemical validity**.

Previous studies have often focused too narrowly on geometric complementarity, overlooking other essential requirements a drug-like molecule must fulfill. We introduce the term *drug potential* to describe the overall suitability of a molecule as a drug candidate, encompassing not only binding affinity but also synthetic accessibility, chemical stability, pharmacokinetics (absorption, distribution, metabolism, and excretion), and safety. These properties are intrinsically determined by molecular structure and collectively influence whether a molecule can reach its intended biological target and survive the drug development process.

The term *drug-likeness* is widely used in medicinal chemistry to reflect the multidimensional suitability of a molecule as a drug candidate. Drug discovery and development (DDD), however, are deeply influenced by human expertise—including implicit preferences for molecular scaffolds and nuanced, often tacit, domain knowledge that is difficult to formalize or quantify. As a result, even experienced medicinal chemists struggle to define or approximate the true probability function $p(\text{drug})$, which represents the likelihood that a molecule will become a viable therapeutic candidate. However, the machine learning community often oversimplifies *drug-likeness* to metrics such as QED or Lipinski's Rule of Five, which capture only a narrow range of basic physicochemical properties. This simplification overlooks critical factors such as oral bioavailability, metabolic stability, and toxicity risks (e.g., hERG liability).

At a more fundamental level, a molecule must be *chemically valid*, meaning it adheres to basic chemical rules such as proper valence and atom types. However, we observe that many model-generated molecules—while technically valid—contain rare or unstable structural substructures that would be flagged by human medicinal chemists. These structures are neither common nor practically accessible and thus fall outside the bounds of what is typically accepted in pharmaceutical research. Despite the central importance of this distinction, prior work has not proposed an effective metric to differentiate between chemically plausible structures and those that are formally valid but unrealistic. To fill this gap, we propose MRR, a rule-based metric that reflects medicinal chemistry heuristics. It identifies implausible features such as unstable ring systems and uncommon conjugation patterns, offering an interpretable and practical means of identifying unrealistic model outputs.

By explicitly defining these concepts and introducing MRR, we aim to guide molecular generation efforts toward pharmaceutically meaningful directions, bridging the gap between computational outputs and real-world drug development feasibility.

## B Limitations

One limitation of CIDD is its dependence on pretrained LLMs, which may occasionally introduce hallucinations in underexplored chemical regions.

## C Detailed Prompts and Responses for LEDD

In this section, we present the detailed workflow of the CIDD framework, including the prompts and example responses for each module.

Figure 5 illustrates the complete drug design pipeline. The Interaction Module first identifies key fragments within the supporting molecule that interact with the protein pocket. This information is then utilized by the Design Module, which devises strategies to replace uncommon or unfavorable fragments while preserving crucial interactions. Once a new molecule is designed, the Evaluation

Phase within the Design Module assesses its viability. Finally, the Reflection Module analyzes the design process and outcomes, highlighting both strengths and areas for improvement.

Figure 6 presents the prompt and example response for the Interaction Analysis Module.

Figures 7 and 8 display the prompt and example response for the Design Module.

Figures 9, 10, and 11 illustrate the prompt and example responses for the Reflection Module.

Figures 12 and 13 show the prompt and example response for the Selection Module.

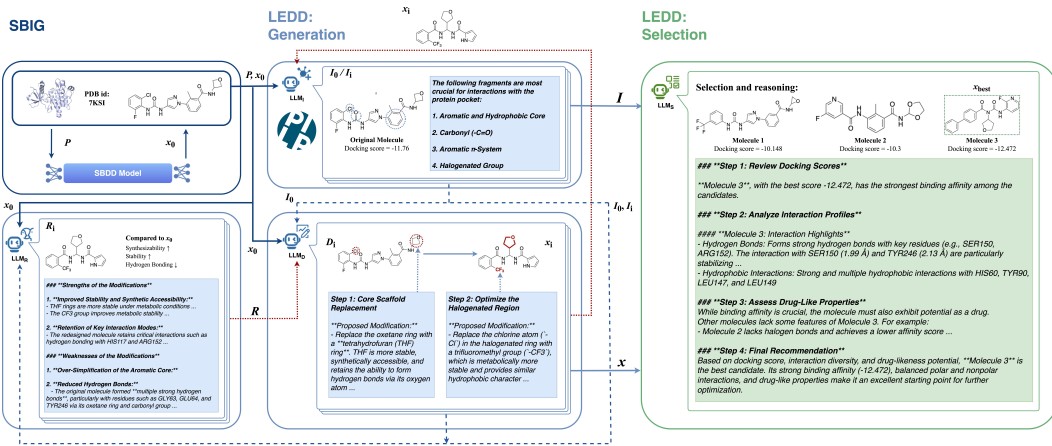

Figure 5: Workflow of CIDD framework

## Interaction analysis

*The interaction analysis results are as follows:* **[PLIP(Docking($x_0$, $P$))]**

*The smiles string of the molecule is:* **[$x_0$]**

**Query**

*[Important]* Using the information from the interaction analysis and the fragment labels, based on the coordinates matching, you need to identify the key fragment on the molecule that conduct the interaction with the protein pocket.
*[Important]* The format should be, which fragment can interact with which residue in the protein pocket. For example: The fragment 'ethanol' can interact with the residue 'ASP123' in the protein pocket.
*[Important]* It should be at fragment level, instead of atom level. You need use the information from the fragment labels to identify which atom belongs to which fragment.

**LLM_I**

**Output [$I_0$]:**

### **Key Fragments of Importance**
Based on the analysis, the following fragments are most crucial for interactions with the protein pocket:

**1. Aromatic and Hydrophobic Core:**
 - `[C]C1=C[C]=[C][C]=C1` (Atoms: 5-11)
 - Critical for hydrophobic interactions with ILE113, LEU142, and LEU147. This fragment anchors the molecule in the hydrophobic pocket.

**2. Carbonyl (-C=O):**
 - Atoms `12, 13`
 - Vital for hydrogen bonds, particularly with GLY63 and GLU64. This fragment contributes to the specificity of the interaction.

**3. Oxetane Ring:**
 - `[C]1[C][C]O1` (Atoms: 15-18)
 - Key for hydrogen bonding with TYR246, providing excellent specificity and strength.

**4. Aromatic π-System:**
 - `[C]1=NN[C]=C1` (Atoms: 0-4)
 - Engaged in π-stacking with TYR90, stabilizing the ligand in the binding pocket.

**5. Halogenated Group:**
 - `FC1=CC(Cl)=[C][C]=[C]1` (Atoms: 23-30)
 - Responsible for halogen bonding with PHE61, contributing to unique and specific interactions.

Figure 6: Interaction analysis module

[Instruction]

*This is the original molecule:* $[x_0]$

*This is the interaction analysis:* $[I_0]$

*Based on the interaction analysis, design modifications to the original molecule. Your job is scaffold hopping: change the core scaffold structure while retaining key pharmacophores.*

[Design Objectives]
*1. The modified molecule should be more stable and easier to synthesize than the original molecule. The new fragments should be common, stable and not weird.*
*2. The modified molecule should retain key properties of the original molecule, such as shape, size, and functionality.*
*3. The modified molecule should contain the key interaction types with the same residues as the original molecule.*
*4. The original molecule may contain uncommon or unstable fragments. Identify these fragments and replace them with more common and stable counterparts, ensuring that the modifications retain key properties of the original molecule, such as shape, size, and functionality. For instance, you might replace cyclohexadiene with benzene.*
*5. The modified molecule should be more like a drug candidate than the original molecule.*

[Design Requirements]
*1. Do not modify the molecule too much.*
*2. Please only give the modification steps, do not generate smiles by yourself. We will use another agent to generate the new molecule based on your design.*

[Experience Summary]
*We have done some modifications before. You can learn from the previous designs and reflections to make better modifications.*
*Here is the summary of the previous designs and reflections:* $[R]$

*Now, give us the modification suggestions and requirements.*

**Query**

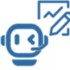
**LLM_D**

**Output [$D_1$]:**

### **Step 1: Core Scaffold Replacement**
**Unstable or Uncommon Scaffold:**
- The original scaffold contains an oxetane ring (`[C]1[C][C]O1`), which, while beneficial for solubility and hydrogen bonding, can be chemically unstable under metabolic conditions and is synthetically challenging.

**Proposed Modification:**
- Replace the oxetane ring with a **tetrahydrofuran (THF) ring**. THF is more stable, synthetically accessible, and retains the ability to form hydrogen bonds via its oxygen atom. Importantly, it maintains a similar size, shape, and polarity.

### **Step 2: Optimize the Halogenated Region**
**Uncommon or Unstable Fragment:**
- The halogenated aromatic fragment (`FC1=CC(Cl)=[C][C]=[C]1`) can contribute to metabolic instability or toxicity (especially with chlorine) in some cases. The position and type of halogenation also determine interaction strength.

**Proposed Modification:**
- Replace the chlorine atom (`-Cl`) in the halogenated ring with a trifluoromethyl group (`-CF3`), which is metabolically more stable and provides similar hydrophobic character and electronic properties to maintain halogen-bonding interactions. The fluorine atom can remain unaltered.

### **Step 3: Revise the Aromatic Core**
**Uncommon or Unstable Fragment:**
- The core ring `[C]C1=C[C]=[C][C]=C1` is a polycyclic conjugated system prone to oxidative metabolism. While it provides hydrophobicity and anchors the molecule, replacing it with a more common aromatic fragment (e.g., phenyl) can reduce metabolic liabilities without compromising interactions.

**Proposed Modification:**
- Replace `[C]C1=C[C]=[C][C]=C1` with **a phenyl ring (benzene)**. Phenyl rings retain hydrophobicity and size while reducing the likelihood of oxidation and increasing synthetic accessibility.

Figure 7: Design Module

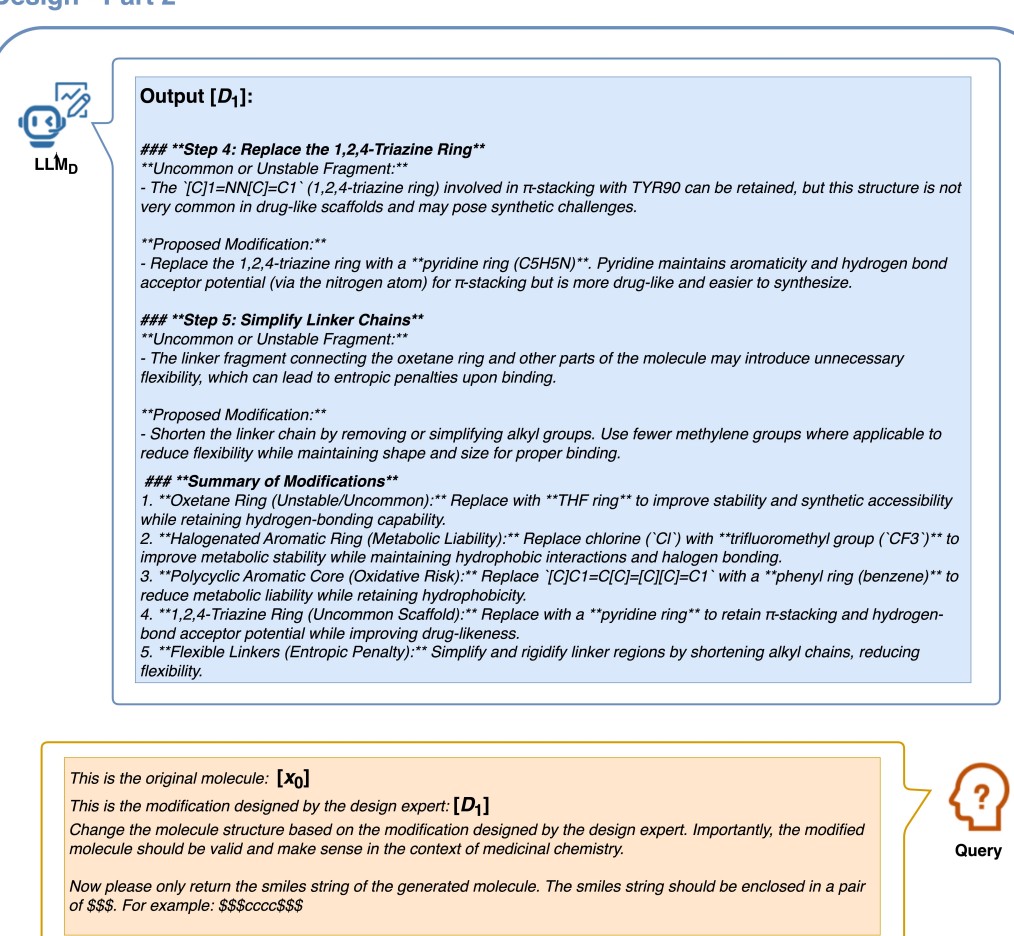

**Output [$D_1$]:**

### **Step 4: Replace the 1,2,4-Triazine Ring**
**Uncommon or Unstable Fragment:**
- The `[C]1=NN[C]=C1` (1,2,4-triazine ring) involved in π-stacking with TYR90 can be retained, but this structure is not very common in drug-like scaffolds and may pose synthetic challenges.

**Proposed Modification:**
- Replace the 1,2,4-triazine ring with a **pyridine ring (C5H5N)**. Pyridine maintains aromaticity and hydrogen bond acceptor potential (via the nitrogen atom) for π-stacking but is more drug-like and easier to synthesize.

### **Step 5: Simplify Linker Chains**
**Uncommon or Unstable Fragment:**
- The linker fragment connecting the oxetane ring and other parts of the molecule may introduce unnecessary flexibility, which can lead to entropic penalties upon binding.

**Proposed Modification:**
- Shorten the linker chain by removing or simplifying alkyl groups. Use fewer methylene groups where applicable to reduce flexibility while maintaining shape and size for proper binding.

### **Summary of Modifications**
1. **Oxetane Ring (Unstable/Uncommon):** Replace with **THF ring** to improve stability and synthetic accessibility while retaining hydrogen-bonding capability.
2. **Halogenated Aromatic Ring (Metabolic Liability):** Replace chlorine (`Cl`) with **trifluoromethyl group (`CF3`)** to improve metabolic stability while maintaining hydrophobic interactions and halogen bonding.
3. **Polycyclic Aromatic Core (Oxidative Risk):** Replace `[C]C1=C[C]=[C][C]=C1` with a **phenyl ring (benzene)** to reduce metabolic liability while retaining hydrophobicity.
4. **1,2,4-Triazine Ring (Uncommon Scaffold):** Replace with a **pyridine ring** to retain π-stacking and hydrogen-bond acceptor potential while improving drug-likeness.
5. **Flexible Linkers (Entropic Penalty):** Simplify and rigidify linker regions by shortening alkyl chains, reducing flexibility.

---

This is the original molecule: **[$x_0$]**

This is the modification designed by the design expert: **[$D_1$]**

Change the molecule structure based on the modification designed by the design expert. Importantly, the modified molecule should be valid and make sense in the context of medicinal chemistry.

Now please only return the smiles string of the generated molecule. The smiles string should be enclosed in a pair of $$$. For example: $$$cccc$$$

**Query**

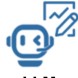
**LLM$_D$**

**Output [$x_1$]:**

$$$C1=CC(C(F)(F)F)=C(C(=O)NC(C2CCOC2)NC(=O)C3=CC=CN3)C=C1$$$

Figure 8: Design Module

## Reflection - Part 1

**[Instruction]**
================================================================================
This is the original molecule structure: $[x_0]$

This is the interaction analysis of this molecule and the protein pocket: $[I_0]$
================================================================================
This is the modification design generated by you: $[D_1]$

This is the molecule structure designed based on your modification: $[x_1]$

This is the interaction analysis generated by interaction export agent: $[I_1]$

Previously, we asked you to modify a molecule to get a new molecule that binds better to the protein pocket. We analyzed the binding of the new molecule to the protein pocket and put it together with the binding before the modification for comparison.
Based on these analyses, you need to reflect on the previous modification and provide feedback on the design. Besides, we hope you can find out which fragments inside the structure of these two molecules are useful for binding to the protein pocket.
================================================================================
Please use critical thinking to analyze, pointing out both the good and the bad points. Your reflection is:

**Query**

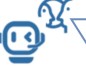

**LLM_R**

**Output $[R_1]$:**

### **Strengths of the Modifications**
1. **Improved Stability and Synthetic Accessibility:**
   - Replacing the **oxetane ring** with a **tetrahydrofuran (THF) ring** is a commendable adjustment. THF rings are more stable under metabolic conditions, while retaining key hydrogen-bonding interactions and similar steric bulk. This was a sensible change that aligns well with drug-likeness principles.
   - Removing the **chlorine atom** in favor of a **trifluoromethyl (CF3) group** was another thoughtful modification. The CF3 group improves metabolic stability and reduces potential toxicity from reactive halogens, while maintaining the original hydrophobicity and shape.

2. **Retention of Key Interaction Modes:**
   - The redesigned molecule retains critical interactions such as hydrogen bonding with HIS117 and ARG152, hydrophobic interactions with HIS60, LEU147, and PHE188, and π-stacking with TYR90. These interactions are crucial for binding affinity and ensure the molecule continues to engage the protein pocket effectively.
   - The **pyridine ring** (replacing the less common 1,2,4-triazine ring) aligns better with drug-likeness rules while maintaining hydrogen bonding and hydrophobic capabilities. This substitution strengthens the molecule's pharmacokinetic prospects.

3. **Refinement of Scaffolds and Simplification:**
   - Simplifying the polycyclic aromatic core into a **benzene ring** was a reasonable modification to reduce oxidation-prone elements. This ensures the molecule is more stable and synthetically tractable without losing significant hydrophobic interactions.

### **Weaknesses of the Modifications**
1. **Reduction in Docking Score:**
   - The docking score decreased from **-11.76 (original molecule)** to **-9.215 (modified molecule)**, indicating a clear drop in binding affinity. While the modified molecule retains key interactions, some critical binding contributions were inadvertently weakened during redesign. The TEF reduction suggests a loss of key complementary contacts between the ligand and protein pocket.

2. **Over-Simplification of the Aromatic Core:**
   - Replacing the conjugated aromatic system (`[C]C1=C[C]=[C][C]=C1`) with a simple **benzene ring** reduced the hydrophobic surface area and π-electron system available for interactions. Consequently, hydrophobic interactions with residues like TYR90, ILE113, and LEU142 were diminished, and the molecule lost some of its ability to stabilize key π-interactions within the pocket.

3. **Unutilized Potential of the Trifluoromethyl Group:**
   - While the trifluoromethyl (CF3) group improved metabolic stability, it did not directly contribute to binding interactions within the protein pocket. This functional group occupies space but does not engage with any specific residues, effectively adding to molecular weight without enhancing binding. A better-placed or functionally active replacement could have yielded greater interaction strength.

Figure 9: Reflection Module

## Reflection - Part 2

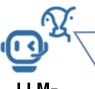
**LLM$_R$**

**Output [$R_1$]:**

4. **Reduced Hydrogen Bonds:**
   - The original molecule formed **multiple strong hydrogen bonds**, particularly with residues such as GLY63, GLU64, and TYR246 via its oxetane ring and carbonyl group. These hydrogen bonds were either weakened or lost in the redesign, with new hydrogen bonds (forming with HIS117 and ARG152) failing to fully compensate for this depletion.

5. **Flexibility in the THF Ring:**
   - While the THF ring is more stable than the oxetane ring, it likely introduces greater flexibility due to its sp3-hybridized carbons. This flexibility may lead to entropic penalties upon binding, making the molecule less efficient in achieving a preorganized, bioactive conformation.

6. **Over-Reliance on Hydrophobic Interactions:**
   - Similar to the original molecule, the modified structure still relies heavily on **hydrophobic interactions** for binding (e.g., HIS60, TYR90, LEU147). Hydrophobic interactions are less specific than polar or ionic interactions, which are critical for selectivity and affinity. By not introducing additional polar groups, the redesign missed an opportunity to improve overall binding specificity.

### **Comparison of Binding Fragments**
#### **Original Molecule:**
1. **Oxetane Ring (Lost in the Redesign):**
   - Forms key hydrogen bonds with TYR246 in the binding pocket, contributing significantly to specific and strong ligand binding. Its removal weakened overall binding affinity.
2. **Conjugated Hydrophobic Core (`[C]C1=C[C]=[C][C]=C1`):**
   - Provides extensive hydrophobic and π-stacking interactions with residues such as ILE113, LEU142, and TYR90. Replacing this rich aromatic structure with a benzene ring reduced the hydrophobic and π-electron contribution to binding.
3. **Halogenated Aromatic Ring (`FC1=CC(Cl)=[C][C]=[C]1`):**
   - Engages the protein via halogen bonding with PHE61, adding specificity to binding interactions. Replacing chlorine with CF3 abolished some halogen bonds, weakening protein-ligand complementarity.

#### **Modified Molecule:**
1. **THF Ring (Introduced):**
   - Retains the ability to form hydrogen bonds (e.g., with ARG152). However, its introduction has caused greater flexibility and entropic penalties that may offset its stabilizing contributions.
2. **Trifluoromethyl Group (CF3, Introduced):**
   - Metabolically stable and hydrophobic but does not directly interact with the protein. Its placement on the aromatic ring adds molecular weight but no functional enhancement to binding.
3. **Pyridine Ring (Introduced):**
   - Maintains hydrogen-bonding with HIS117 and interacts hydrophobically with HIS60, performing well as a replacement for 1,2,4-triazine. Though not as inherently diverse in interaction opportunities as its predecessor, it improves the drug-likeness of the molecule.
4. **Benzene Ring (Simplified):**
   - While it retains some hydrophobic interaction capability, this fragment no longer engages residues like TYR90 or PHE188 as robustly as the original conjugated aromatic system. This reduced binding affinity.

Figure 10: Reflection Module

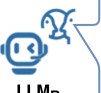

**LLMR**

**Output [$R_1$]:**

### **Key Fragments Retained or Lost for Binding**
**Critical Retained Fragments:**
1. **Pyridine Ring** for π-stacking and hydrogen bonding with TYR90 and HIS117.
2. **Hydrogen Bonding Carbonyl Group** interacting with HIS117 and ARG152.

**Critical Fragments Lost:**
1. **Oxetane Ring** contributed unusually strong hydrogen bonds to TYR246, which were lost in the redesign.
2. **Conjugated Hydrophobic Core** ensured multisite hydrophobic interactions and π-stacking with residues like LEU142 and TYR90.
3. **Halogenated Aromatic Ring** provided halogen bonds with PHE61, contributing unique specificity lost in the redesign.

### **Suggestions for Improvement**
1. **Reintroduce Hydrogen Bonding Groups:**
   - Add back polar fragments (e.g., hydroxyl, amide groups) to increase hydrogen-bonding capacity with residues like TYR246, GLY63, and GLU64.

2. **Revisit the Aromatic Core:**
   - Instead of oversimplifying the hydrophobic core to benzene, consider a polycyclic aromatic or heterocyclic system to balance hydrophobicity and metabolic stability.

3. **Optimize Trifluoromethyl Group Placement:**
   - Move or replace the CF3 group with a fragment that directly engages residues in the protein pocket (e.g., a difluorophenyl or hydrophilic substituent).

4. **Rigidify the THF Ring:**
   - Lock the THF ring into a constrained system (e.g., fused bicyclic rings) to reduce flexibility and entropic penalties.

### **Conclusion**
The modifications resulted in a molecule that is synthetically and metabolically improved but less effective in binding to the protein pocket, as indicated by the diminished docking score. While retaining important interactions, the redesign lost too many advantageous fragments (e.g., oxetane, conjugated aromatic core). Future optimizations should aim to restore lost interactions, improve hydrogen bonding, and balance hydrophobicity with polar contacts for stronger and more specific binding.

Figure 11: Reflection Module

*[Instruction]*
*Given a protein pocket and a number of different molecules, I need you to find a molecule that is best suited to bind this protein pocket. We will provide you with interaction reports for each molecule and protein pocket and ask you to make choices based on this information. You should consider both the binding analysis and whether the molecule has potential to be a real drug. You should output the reason and the smiles string of the chosen molecule. The smiles string should be enclosed in a pair of $$$. For example: ...(The reason) The chosen smiles is: $$$CCO$$$*

*This is the molecule structure:* $[x_n]$

*This is the interaction analysis of this molecule and the protein pocket:* $[I_n]$
================================================================================
*This is the molecule structure:* $[x_m]$

*This is the interaction analysis of this molecule and the protein pocket:* $[I_m]$
================================================================================
*This is the molecule structure:* $[x_p]$

*This is the interaction analysis of this molecule and the protein pocket:* $[I_p]$
================================================================================
*Please think step by step:*

**Query**

**LLM$_S$**

***The chosen molecule is:***
***O=C(C1=CC=C(C=C1)C2=CC=CC=C2)N(C3CCOC3)C(NC4=C(F)N=CC=C4)=O***
*The docking score of the chosen molecule is: -12.472*
*The reason for choosing the molecule is: To select the best molecule for binding to the protein pocket, we will step through the following considerations:*

*### **Step 1: Review Docking Scores***
*The docking score is a key metric indicating the binding affinity between a molecule and the protein pocket. Lower docking scores (more negative values) suggest stronger binding. The docking scores for the respective molecules are as follows:*
*1. Molecule 1: -10.148*
*2. Molecule 2: -10.3*
*3. Molecule 3: -12.472*

*From this, **Molecule 3**, with the best score of **-12.472**, has the strongest binding affinity among the candidates.*

*### **Step 2: Analyze Interaction Profiles***
*Building off the docking scores, we now consider the rich interaction profiles of each molecule. A molecule that forms diverse interactions will often bind more robustly and specifically to the protein pocket.*

*#### **Molecule 3: Interaction Highlights***
*- **Hydrophobic Interactions:** Strong and multiple hydrophobic interactions with HIS60, TYR90, LEU147, and LEU149, anchored by the aromatic benzene ring.*
*- **Hydrogen Bonds:** Forms strong hydrogen bonds with key residues (e.g., SER150, ARG152). The interaction with SER150 (1.99 Å) and TYR246 (2.13 Å) are particularly stabilizing.*
*- **π-Stacking:** Strong π-stacking with TYR90 (T-shaped geometry).*
*- **π-Cation Interactions:** Robust π-cation interaction with ARG152 (3.85 Å), stabilized further by ideal offset alignment (1.32 Å).*
*- **Halogen Bonds:** Two halogen bonds with GLU189 (3.84 Å) and ASN237 (3.40 Å) provide an additional layer of interaction, making this molecule unique in its interaction diversity.*

Figure 12: Selection Module

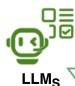

LLMₛ

*Key Strengths:*
- *The interaction diversity and strength of Molecule 3 far surpass other molecules, leveraging hydrophobic forces, hydrogen bonding, π-stacking, π-cation interactions, and halogen bonding.*
- *The molecule effectively anchors itself to both polar and nonpolar regions, ensuring specificity for the protein pocket.*

### *Step 3: Assess Drug-Like Properties*
*While binding affinity is crucial, the molecule must also exhibit potential as a drug. Properties relevant to drug-likeness include:*
- *Hydrophobic Components:* *Ensure solubility and stability for membrane diffusion.*
- *Hydrogen Bonding Groups:* *Contribute to specificity and bioactivity.*
- *Halogen Incorporation:* *Enhances membrane permeability and metabolic stability.*
- *Compact, Non-Bulky Structure:* *Ensures the molecule fits within the protein pocket without steric clashes.*

*For Molecule 3:*
- *The combination of aromatic rings, amide groups, and fluorine atoms supports balanced hydrophobic and polar properties, increasing stability and bioavailability.*
- *The inclusion of halogens enhances drug-likeness further.*

*Other molecules lack some features of Molecule 3. For example:*
- *Molecule 2 lacks halogen bonds and achieves a lower affinity score.*
- *Molecule 1 have good binding but lack the breadth of interactions seen with Molecule 3.*

### *Step 4: Final Recommendation*
*Based on docking score, interaction diversity, and drug-likeness potential, **Molecule 3** is the best candidate. Its strong binding affinity (-12.472), balanced polar and nonpolar interactions, and drug-like properties make it an excellent starting point for further optimization.*

### *Chosen Molecule*
*Reason:* *Molecule 3 exhibits the highest docking score, with diverse interaction types (hydrophobic, hydrogen bonding, π-stacking, π-cation, and halogen bonding) that secure it in critical regions of the protein pocket. Additionally, it offers promising drug-like features such as halogen incorporation and a well-optimized molecular structure.*

*Selected SMILES String:*
*$$$O=C(C1=CC=C(C=C1)C2=CC=CC=C2)N(C3CCOC3)C(NC4=C(F)N=CC=C4)=O$$$*

Figure 13: Selection Module

# D  Algorithm for MRR and AUR

The complete calculation process for assessing the reasonability of a molecule is outlined in Algorithm 1.

---

**Algorithm 1:** Evaluation of Molecular Reasonability

---

**Input:** Molecule object (*mol*)
**Output:** Molecular Reasonability (*MRR*) and Atom Unreasonable Ratio (*AUR*)
**Step 1: Detect Carbonyl and Imine Group Carbons**
Initialize an empty list for *carbonyl/imine carbons*.
**foreach** *bond in mol* **do**
  **if** *bond is double and one atom is carbon, the other is oxygen or nitrogen* **then**
    Record the carbon atom in *carbonyl/imine groups*.

**Step 2: Identification of Ring Systems**
Identify all ring structures and their corresponding atom indices within *mol*.
Calculate the number of atoms in each ring.
**foreach** *ring in the molecule* **do**
  **if** *the ring shares one or more atoms with another ring* **then**
    Group the connected rings into a single *ring system*.

**Step 3: Evaluation of Molecular Reasonability**
Exclude any atoms previously identified as part of carbonyl or imine groups.
Classify the remaining carbon atoms in each ring system as follows:
  • *$sp^2$ hybridized*: Aromatic or unsaturated carbons.
  • *Non-$sp^2$ hybridized*: Saturated carbons.

**foreach** *ring system in the ring systems* **do**
  **if** *the ring system contains multiple rings and all carbon atoms are non-$sp^2$* **then**
    Mark the molecule as unreasonable.
    Add the atoms to the unreasonable atom list.

**foreach** *ring system in the remaining ring systems* **do**
  **foreach** *ring in the ring system* **do**
    **if** *all carbon atoms within the ring are consistent in hybridization (either all $sp^2$ or all non-$sp^2$)* **then**
      Mark the ring as reasonable.
    **else**
      Add the ring to the *remaining ring list*.

**while** *the remaining ring list is not empty* **do**
  **foreach** *ring in the remaining ring list* **do**
    Exclude atoms that have already been classified as reasonable.
    **if** *all remaining carbon atoms are consistent in hybridization (either all $sp^2$ or all non-$sp^2$)* **then**
      Mark the ring as reasonable.

  **if** *no new reasonable rings are identified* **then**
    Mark the molecule as unreasonable.
    Add the carbon atoms in the remaining rings to the unreasonable atom list.
    **Exit the loop.**

Calculate *AUR* as the ratio of unreasonable atom count to the total ring atom count.
**Return** *MRR* and *AUR*.

---

# E   QikProp properties

The full set of properties used for the QikProp pass ratio analysis is presented in Table 3.

The QikProp filter applied in the main text incorporates a comprehensive range of criteria provided by QikProp, including "#stars", "#amine", "#amidine", "#acid", "#amide", "#rotor", "#rtvFG", "mol_MW", "dipole", "SASA", "FOSA", "FISA", "PISA", "WPSA", "volume", "donorHB", "acptHB", "dip$^2/V$", "ACxDN·5/SA", "glob", "QPpolrz", "QPlogPC16", "QPlogPoct", "QPlogPw", "QPlogPo/w", "QPlogS", "CIQPlogS", "QPPCaco", "QPlogBB", "QPPMDCK", "QPlogKp", "IP(eV)", "EA(eV)", "#metab", "QPlogKhsa", "PercentHumanOralAbsorption", "SAFluorine", "SAamideO", "PSA", "#NandO", and "RuleOfThree".

Table 3: QikProp Properties and Descriptors

| Property or Descriptor | Description | Range or Recommended Values |
|---|---|---|
| Molecule name | The molecule's identifier derived from the title line in the input structure file. If no title is provided, the file name is used. | |
| #stars | Count of descriptors or properties falling outside the 95% range for known drugs. A higher count indicates reduced drug-likeness. | 0 – 5 |
| #amine | Total non-conjugated amine groups present in the molecule. | 0 – 1 |
| #amidine | Number of amidine or guanidine functional groups in the structure. | 0 |
| #acid | Quantity of carboxylic acid groups in the molecule. | 0 – 1 |
| #amide | Count of non-conjugated amide groups. | 0 – 1 |
| #rotor | Number of rotatable bonds that are neither trivial nor sterically hindered. | 0 – 15 |
| #rtvFG | Total reactive functional groups present in the molecule, potentially affecting stability or toxicity. | 0 – 2 |
| mol_MW | Molecular weight of the compound. | 130.0 – 725.0 |
| Dipole | Calculated dipole moment of the molecule in Debye units. | 1.0 – 12.5 |
| SASA | Solvent-accessible surface area (SASA) in square angstroms, measured with a probe of 1.4 Å radius. | 300.0 – 1000.0 |
| FOSA | Hydrophobic part of the SASA, representing saturated carbon and attached hydrogen atoms. | 0.0 – 750.0 |
| FISA | Hydrophilic fraction of the SASA, encompassing polar atoms like nitrogen and oxygen. | 7.0 – 330.0 |
| PISA | SASA component attributable to $\pi$-systems. | 0.0 – 450.0 |
| WPSA | Weakly polar component of the SASA, including atoms like halogens, phosphorus, and sulfur. | 0.0 – 175.0 |
| Volume | Total solvent-accessible volume in cubic angstroms, determined with a 1.4 Å radius probe. | 500.0 – 2000.0 |
| donorHB | Estimated number of hydrogen bonds donated to water in solution. | 0.0 – 6.0 |
| accptHB | Estimated number of hydrogen bonds accepted from water. | 2.0 – 20.0 |
| Dip$^2$/V | Dipole moment squared divided by molecular volume, a key factor in solvation energy. | 0.0 – 0.13 |
| ACxDN$^{0.5}$/SA | Cohesive interaction index in solids based on molecular properties. | 0.0 – 0.05 |
| glob | Descriptor measuring how close the shape of a molecule is to a sphere. | 0.75 – 0.95 |
| QPpolrz | Predicted molecular polarizability in cubic angstroms. | 13.0 – 70.0 |
| QPlogPC16 | Predicted partition coefficient between hexadecane and gas phases. | 4.0 – 18.0 |
| QPlogPoct | Predicted partition coefficient between octanol and gas phases. | 8.0 – 35.0 |
| QPlogPw | Predicted partition coefficient between water and gas phases. | 4.0 – 45.0 |
| QPlogPo/w | Predicted partition coefficient between octanol and water phases. | -2.0 – 6.5 |
| QPlogS | Predicted solubility of the molecule in water (log S, in mol/L). | -6.5 – 0.5 |
| CIQPlogS | Conformation-independent prediction of water solubility (log S). | -6.5 – 0.5 |
| QPPCaco | Predicted permeability through Caco-2 cells, in nm/s. | <25 poor, >500 great |
| QPlogBB | Predicted partition coefficient for brain/blood. | -3.0 – 1.2 |
| QPPMDCK | Predicted permeability through MDCK cells, in nm/s. | <25 poor, >500 great |
| QPlogKp | Predicted skin permeability (log Kp). | -8.0 – -1.0 |
| IP(eV) | Ionization potential calculated using PM3. | 7.9 – 10.5 |
| EA(eV) | Electron affinity calculated using PM3. | -0.9 – 1.7 |
| #metab | Predicted number of possible metabolic reactions. | 1 – 8 |
| QPlogKhsa | Predicted binding affinity to human serum albumin. | -1.5 – 1.5 |
| HumanOralAbsorption | Qualitative assessment of oral absorption: 1 (low), 2 (medium), or 3 (high). | |
| PercentHumanOralAbsorption | Quantitative prediction of oral absorption percentage. | >80% high, <25% poor |
| SAFluorine | Solvent-accessible fluorine surface area. | 0.0 – 100.0 |
| SAamideO | Solvent-accessible surface area of amide oxygen atoms. | 0.0 – 35.0 |
| PSA | Polar surface area, calculated for nitrogen, oxygen, and carbonyl groups. | 7.0 – 200.0 |
| #NandO | Total count of nitrogen and oxygen atoms. | 2 – 15 |
| RuleOfFive | Number of Lipinski's Rule of Five violations. | Max 4 |
| RuleOfThree | Number of Jorgensen's Rule of Three violations. | Max 3 |
| #ringatoms | Count of atoms within molecular rings. | |
| #in34 | Number of atoms in 3- or 4-membered rings. | |
| #in56 | Number of atoms in 5- or 6-membered rings. | |
| #noncon | Number of ring atoms unable to form conjugated aromatic systems. | |
| #nonHatm | Count of heavy (non-hydrogen) atoms in the structure. | |
| Jm | Predicted maximum transdermal transport rate ($\mu$g cm$^{-2}$ hr$^{-1}$). | |

# F   Computing Resource

In this work, we primarily utilize pretrained 3D generative models and large language model (LLM) APIs to conduct our experiments. The 3D model sampling is performed using a single NVIDIA A100 GPU. For the LLM component, we rely on API-based access provided by the service provider, which requires no local computational resources.

# G  More Experiment Results

Based on the different criteria presented in Table 3, we provide additional pass ratio results in Table 4.

Filter 1 is identical to the QikProp filter used in the main text.

Filter 2 removes some non-essential properties and focuses on well-defined physicochemical properties, including "#rtvFG", "QPlogS", "QPlogPo/w", "mol_MW", "dipole", "SASA", "FOSA", "FISA", "IP(eV)", "EA(eV)", "#metab", "PercentHumanOralAbsorption", and "PSA".

Filter 3 assesses molecular compliance with the "RuleOfFive" criterion. However, instead of allowing up to four violations as typically recommended, this filter adopts a stricter definition, considering only molecules that fully comply (i.e., setting the maximum allowable violations to zero).

Table 4: QikProp results for different methods with and without CIDD

| Method | Filter 1 | Filter 2 | Filter 3 |
|---|---|---|---|
| **Pocket2Mol** | | | |
| Original | 29.58% | 51.52% | 89.58% |
| CIDD | 56.97% | 75.64% | 92.24% |
| **TargetDiff** | | | |
| Original | 26.32% | 48.20% | 69.47% |
| CIDD | 53.37% | 75.60% | 81.85% |
| **DecompDiff** | | | |
| Original | 29.04% | 53.96% | 55.14% |
| CIDD | 37.54% | 68.48% | 65.64% |
| **MolCRAFT** | | | |
| Original | 22.37% | 43.52% | 66.45% |
| CIDD | 35.22% | 63.23% | 74.09% |

# H  More cases

More generated molecules from CIDD are presented below. For each case, we display the initial supporting molecule derived from 3D-SBDD models alongside the final designed molecules produced by CIDD.

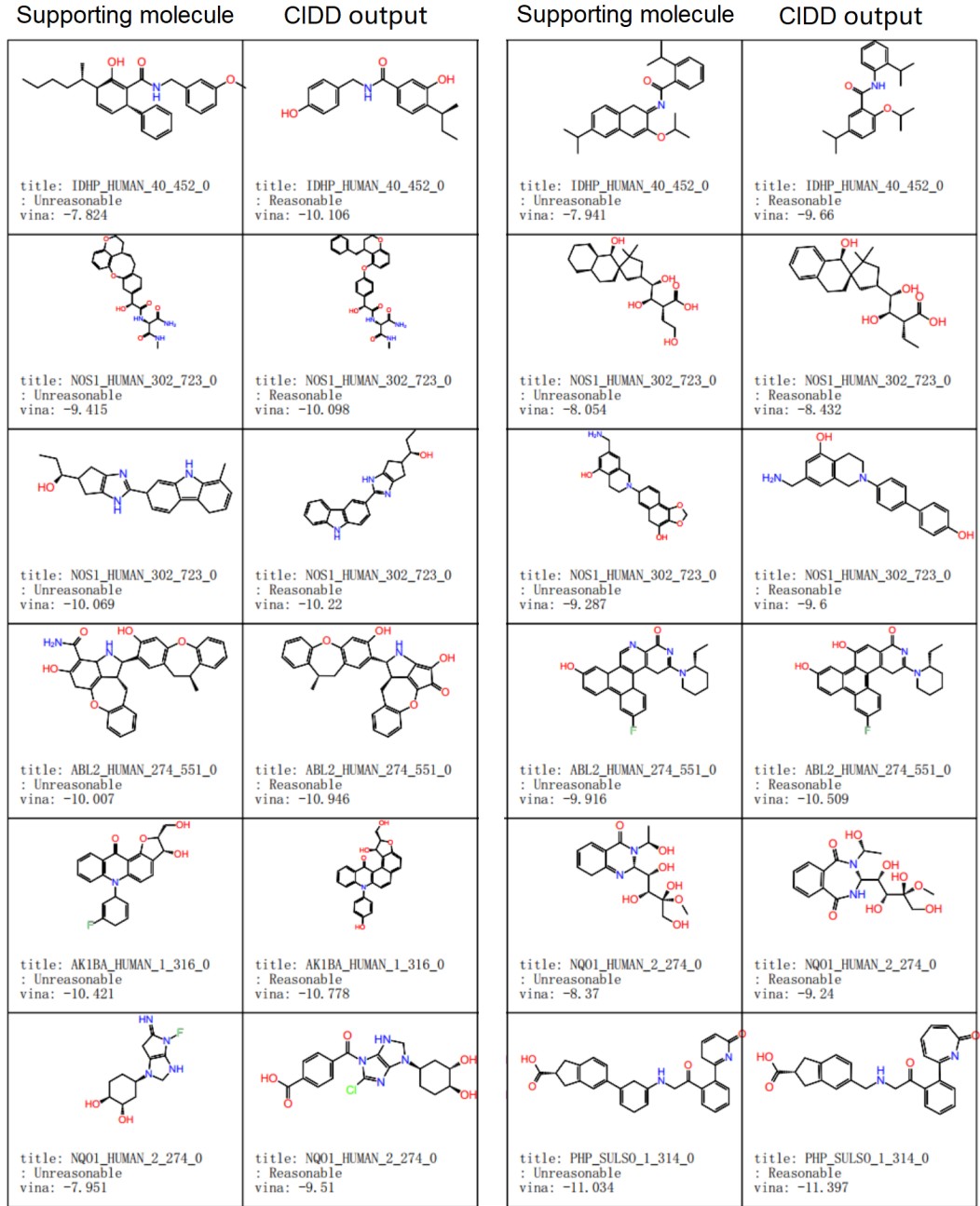

| Supporting molecule | CIDD output | Supporting molecule | CIDD output |
|---|---|---|---|
|  title: PHKG1_RABIT_6_296_ATPsite_0 : Unreasonable vina: -8.548 |  title: PHKG1_RABIT_6_296_ATPsite_0 : Reasonable vina: -8.727 |  title: BGL07_ORYSJ_25_504_0 : Unreasonable vina: -8.041 |  title: BGL07_ORYSJ_25_504_0 : Reasonable vina: -9.949 |
|  title: CD38_HUMAN_44_300_0 : Unreasonable vina: -7.443 |  title: CD38_HUMAN_44_300_0 : Reasonable vina: -8.863 |  title: TNKS2_HUMAN_948_1162_0 : Unreasonable vina: -10.284 |  title: TNKS2_HUMAN_948_1162_0 : Reasonable vina: -10.974 |
|  title: P2Y12_HUMAN_1_342_0 : Unreasonable vina: -10.463 |  title: P2Y12_HUMAN_1_342_0 : Reasonable vina: -10.674 |  title: HMD_METJA_1_358_0 : Unreasonable vina: -9.259 |  title: HMD_METJA_1_358_0 : Reasonable vina: -10.458 |
|  title: SQHC_ALIAD_1_631_0 : Unreasonable vina: -14.278 |  title: SQHC_ALIAD_1_631_0 : Reasonable vina: -17.408 |  title: SQHC_ALIAD_1_631_0 : Unreasonable vina: -13.626 |  title: SQHC_ALIAD_1_631_0 : Reasonable vina: -13.825 |
|  title: BGAT_HUMAN_63_353_0 : Unreasonable vina: -7.891 |  title: BGAT_HUMAN_63_353_0 : Reasonable vina: -9.201 |  title: CHIB_SERMA_1_499_0 : Unreasonable vina: -10.676 |  title: CHIB_SERMA_1_499_0 : Reasonable vina: -13.113 |
|  title: CHIB_SERMA_1_499_0 : Unreasonable vina: -7.115 |  title: CHIB_SERMA_1_499_0 : Reasonable vina: -8.719 |  title: NR1H4_HUMAN_258_486_0 : Unreasonable vina: -10.038 |  title: NR1H4_HUMAN_258_486_0 : Reasonable vina: -11.021 |

| Supporting molecule | CIDD output | Supporting molecule | CIDD output |
|---|---|---|---|
| title: SDIA_ECOLI_1_171_0
: Unreasonable
vina: −11.019 | title: SDIA_ECOLI_1_171_0
: Reasonable
vina: −11.294 | title: SDIA_ECOLI_1_171_0
: Unreasonable
vina: −8.52 | title: SDIA_ECOLI_1_171_0
: Reasonable
vina: −13.937 |
| title: GSTP1_HUMAN_2_210_0
: Unreasonable
vina: −7.591 | title: GSTP1_HUMAN_2_210_0
: Reasonable
vina: −8.725 | title: HDHA_ECOLI_1_255_0
: Unreasonable
vina: −7.709 | title: HDHA_ECOLI_1_255_0
: Reasonable
vina: −8.778 |
| title: HDHA_ECOLI_1_255_0
: Unreasonable
vina: −8.5 | title: HDHA_ECOLI_1_255_0
: Reasonable
vina: −8.515 | title: NOS1_HUMAN_302_723_0
: Unreasonable
vina: −11.943 | title: NOS1_HUMAN_302_723_0
: Reasonable
vina: −12.979 |
| title: IDHP_HUMAN_40_452_0
: Unreasonable
vina: −8.176 | title: IDHP_HUMAN_40_452_0
: Reasonable
vina: −10.774 | title: NR1H4_HUMAN_258_486_0
: Unreasonable
vina: −6.824 | title: NR1H4_HUMAN_258_486_0
: Reasonable
vina: −8.578 |
| title: IDHP_HUMAN_40_452_0
: Unreasonable
vina: −7.605 | title: IDHP_HUMAN_40_452_0
: Reasonable
vina: −11.661 | title: CPXB_BACMB_2_464_0
: Unreasonable
vina: −9.607 | title: CPXB_BACMB_2_464_0
: Reasonable
vina: −10.132 |
| title: AK1BA_HUMAN_1_316_0
: Unreasonable
vina: −9.845 | title: AK1BA_HUMAN_1_316_0
: Reasonable
vina: −9.981 | title: P2Y12_HUMAN_1_342_0
: Unreasonable
vina: −7.75 | title: P2Y12_HUMAN_1_342_0
: Reasonable
vina: −16.253 |
| title: OLIAC_CANSA_1_101_0
: Unreasonable
vina: −7.631 | title: OLIAC_CANSA_1_101_0
: Reasonable
vina: −8.866 | title: SIR3_HUMAN_117_398_0
: Unreasonable
vina: −8.238 | title: SIR3_HUMAN_117_398_0
: Reasonable
vina: −8.331 |

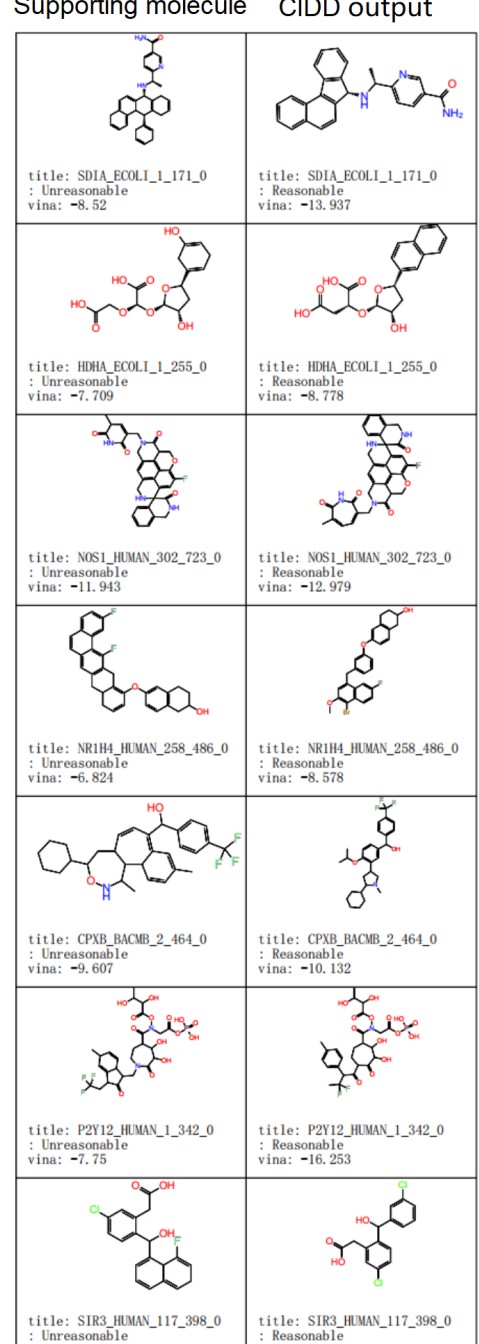

Supporting molecule    CIDD output        Supporting molecule    CIDD output

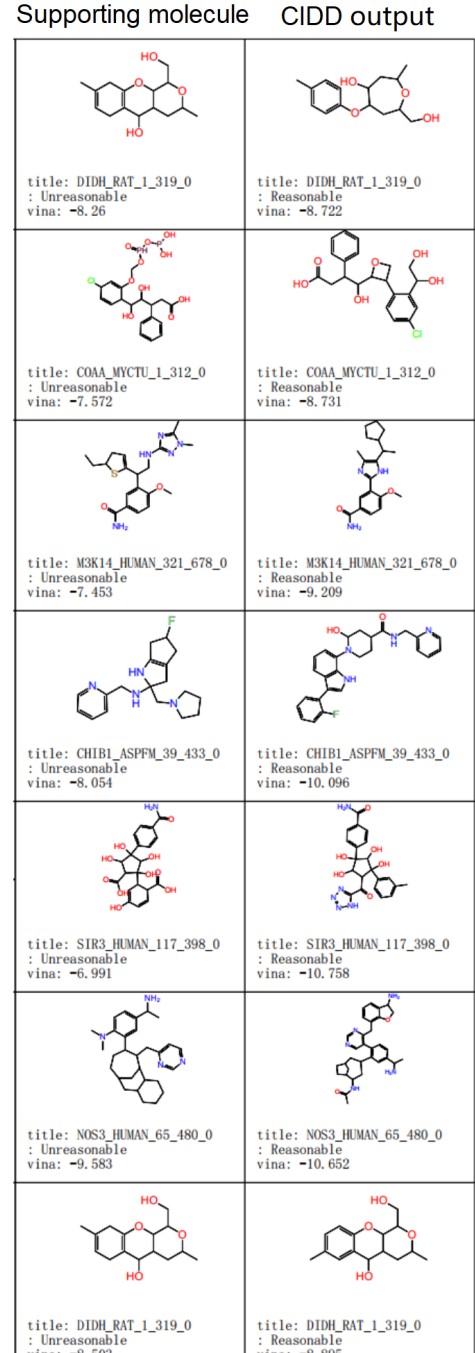

title: MCCF_ECOLX_1_344_0
: Unreasonable
vina: −8.726

title: MCCF_ECOLX_1_344_0
: Reasonable
vina: −8.947

title: DIDH_RAT_1_319_0
: Unreasonable
vina: −8.26

title: DIDH_RAT_1_319_0
: Reasonable
vina: −8.722

title: IDHP_HUMAN_40_452_0
: Unreasonable
vina: −7.796

title: IDHP_HUMAN_40_452_0
: Reasonable
vina: −11.383

title: COAA_MYCTU_1_312_0
: Unreasonable
vina: −7.572

title: COAA_MYCTU_1_312_0
: Reasonable
vina: −8.731

title: AKT1_HUMAN_1_137_0
: Unreasonable
vina: −8.605

title: AKT1_HUMAN_1_137_0
: Reasonable
vina: −9.926

title: M3K14_HUMAN_321_678_0
: Unreasonable
vina: −7.453

title: M3K14_HUMAN_321_678_0
: Reasonable
vina: −9.209

title: HMD_METJA_1_358_0
: Unreasonable
vina: −7.825

title: HMD_METJA_1_358_0
: Reasonable
vina: −8.55

title: CHIB1_ASPFM_39_433_0
: Unreasonable
vina: −8.054

title: CHIB1_ASPFM_39_433_0
: Reasonable
vina: −10.096

title: SIR3_HUMAN_117_398_0
: Unreasonable
vina: −7.93

title: SIR3_HUMAN_117_398_0
: Reasonable
vina: −10.435

title: SIR3_HUMAN_117_398_0
: Unreasonable
vina: −6.991

title: SIR3_HUMAN_117_398_0
: Reasonable
vina: −10.758

title: DYRK2_HUMAN_145_550_0
: Unreasonable
vina: −7.961

title: DYRK2_HUMAN_145_550_0
: Reasonable
vina: −8.399

title: NOS3_HUMAN_65_480_0
: Unreasonable
vina: −9.583

title: NOS3_HUMAN_65_480_0
: Reasonable
vina: −10.652

title: DIDH_RAT_1_319_0
: Unreasonable
vina: −9.165

title: DIDH_RAT_1_319_0
: Reasonable
vina: −10.631

title: DIDH_RAT_1_319_0
: Unreasonable
vina: −8.503

title: DIDH_RAT_1_319_0
: Reasonable
vina: −8.895

Supporting molecule     CIDD output          Supporting molecule     CIDD output

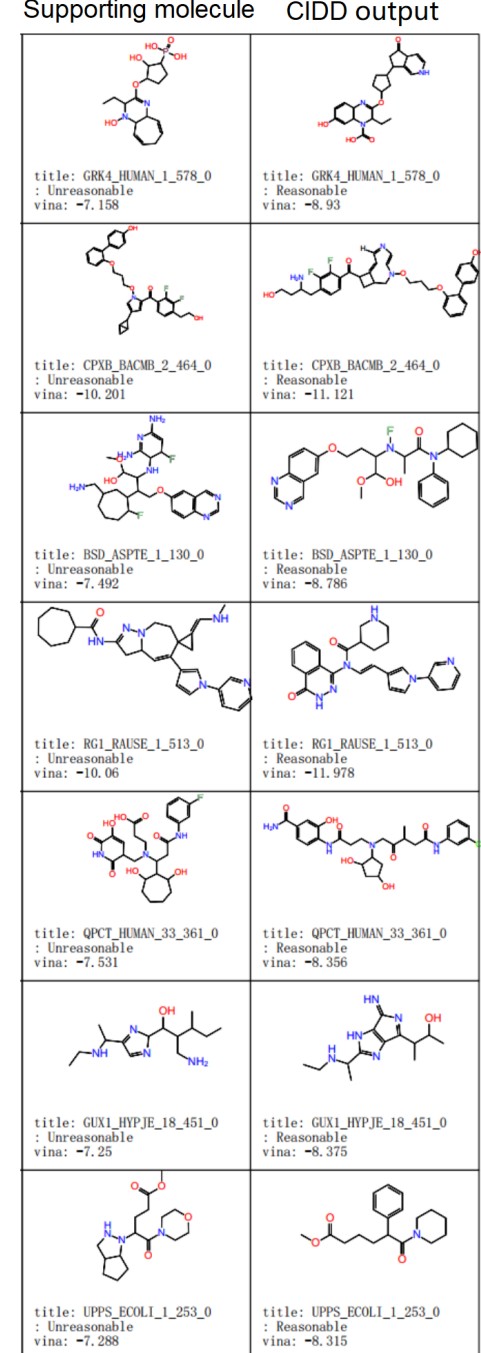

title: BACE2_HUMAN_76_460_0   title: BACE2_HUMAN_76_460_0
: Unreasonable                : Reasonable
vina: −8.586                  vina: −9.854

title: F16P1_HUMAN_1_338_0    title: F16P1_HUMAN_1_338_0
: Unreasonable                : Reasonable
vina: −8.173                  vina: −8.633

title: CAT_ECOLX_1_219_0      title: CAT_ECOLX_1_219_0
: Unreasonable                : Reasonable
vina: −9.838                  vina: −10.164

title: DFPA_LOLVU_2_314_0     title: DFPA_LOLVU_2_314_0
: Unreasonable                : Reasonable
vina: −7.208                  vina: −8.581

title: QPCT_HUMAN_33_361_0    title: QPCT_HUMAN_33_361_0
: Unreasonable                : Reasonable
vina: −8.178                  vina: −9.527

title: KS6A3_HUMAN_41_357_0   title: KS6A3_HUMAN_41_357_0
: Unreasonable                : Reasonable
vina: −8.242                  vina: −8.738

title: AROE_THET8_1_263_0     title: AROE_THET8_1_263_0
: Unreasonable                : Reasonable
vina: −8.741                  vina: −8.794

title: GRK4_HUMAN_1_578_0     title: GRK4_HUMAN_1_578_0
: Unreasonable                : Reasonable
vina: −7.158                  vina: −8.93

title: CPXB_BACMB_2_464_0     title: CPXB_BACMB_2_464_0
: Unreasonable                : Reasonable
vina: −10.201                 vina: −11.121

title: BSD_ASPTE_1_130_0      title: BSD_ASPTE_1_130_0
: Unreasonable                : Reasonable
vina: −7.492                  vina: −8.786

title: RG1_RAUSE_1_513_0      title: RG1_RAUSE_1_513_0
: Unreasonable                : Reasonable
vina: −10.06                  vina: −11.978

title: QPCT_HUMAN_33_361_0    title: QPCT_HUMAN_33_361_0
: Unreasonable                : Reasonable
vina: −7.531                  vina: −8.356

title: GUX1_HYPJE_18_451_0    title: GUX1_HYPJE_18_451_0
: Unreasonable                : Reasonable
vina: −7.25                   vina: −8.375

title: UPPS_ECOLI_1_253_0     title: UPPS_ECOLI_1_253_0
: Unreasonable                : Reasonable
vina: −7.288                  vina: −8.315

| Supporting molecule | CIDD output | Supporting molecule | CIDD output |
|---|---|---|---|

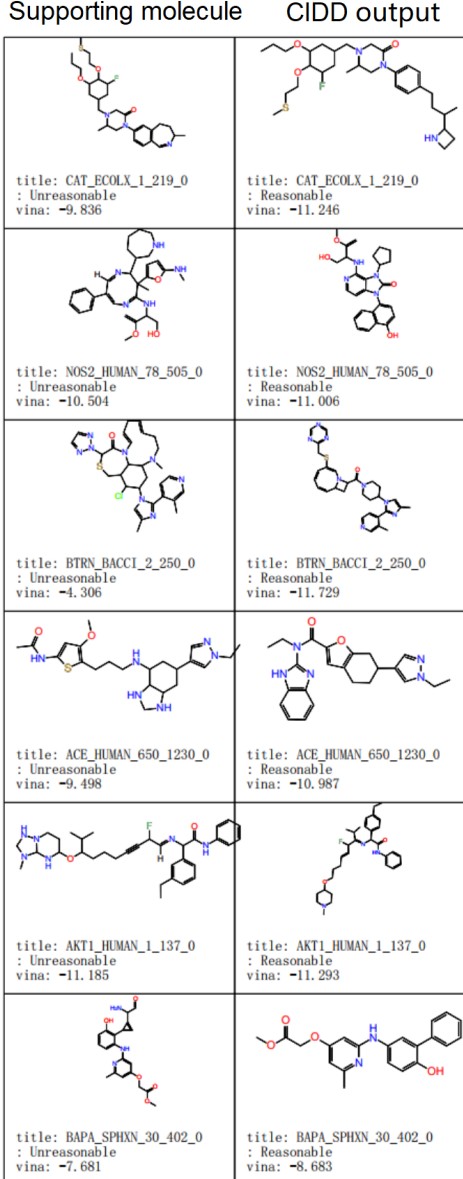
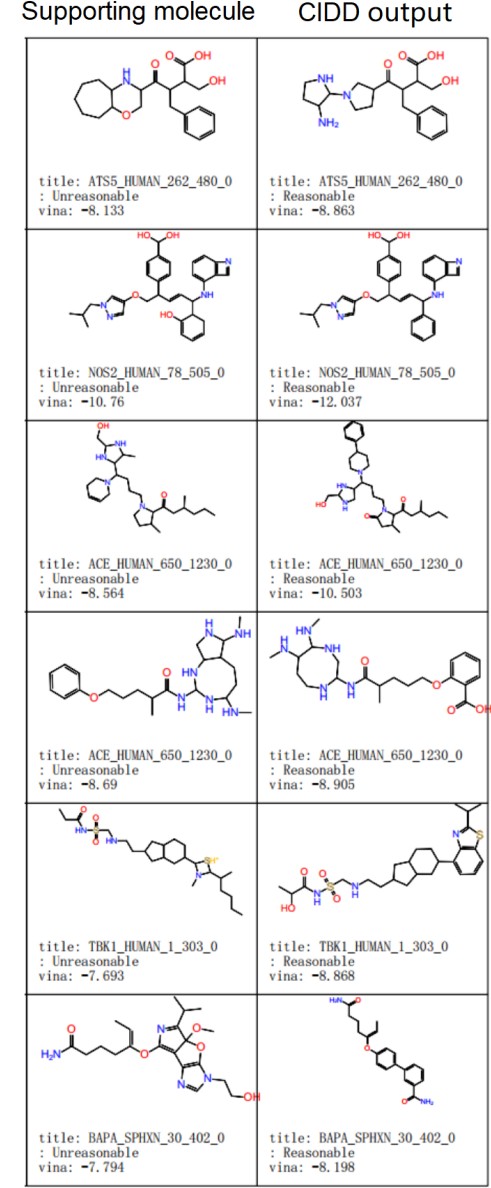

| Supporting molecule | CIDD output |
| --- | --- |

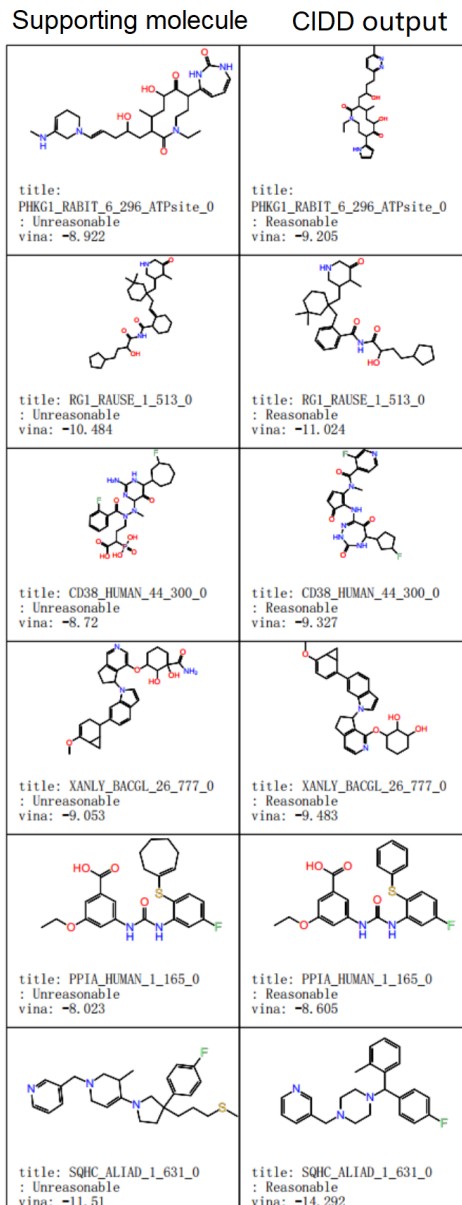

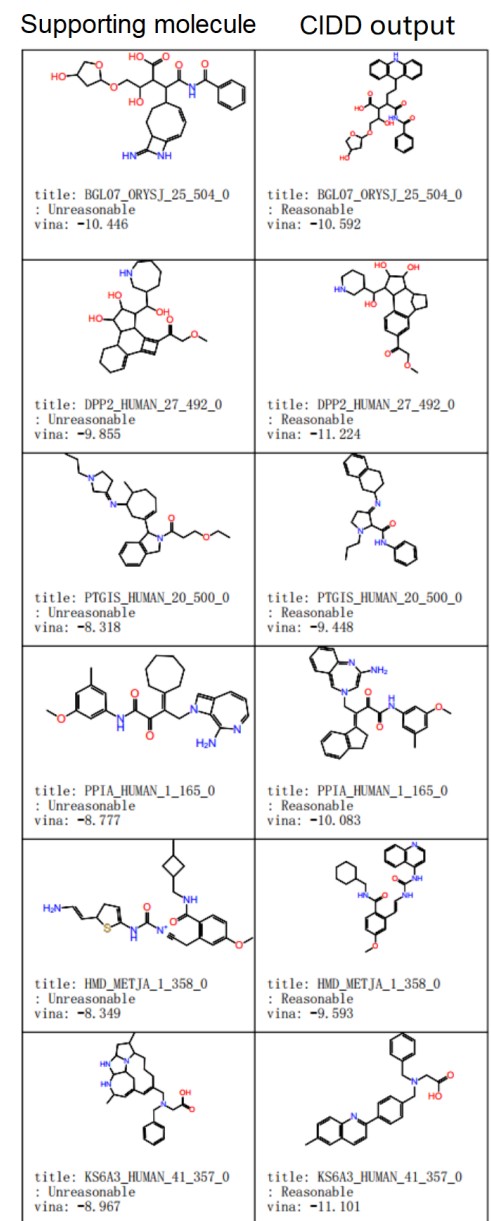

| Supporting molecule | CIDD output | Supporting molecule | CIDD output |
|---|---|---|---|

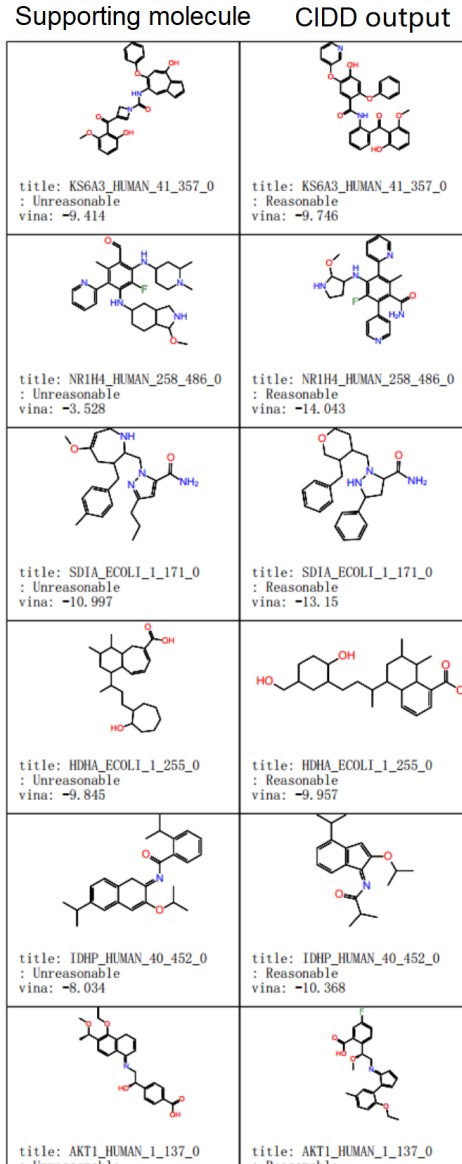

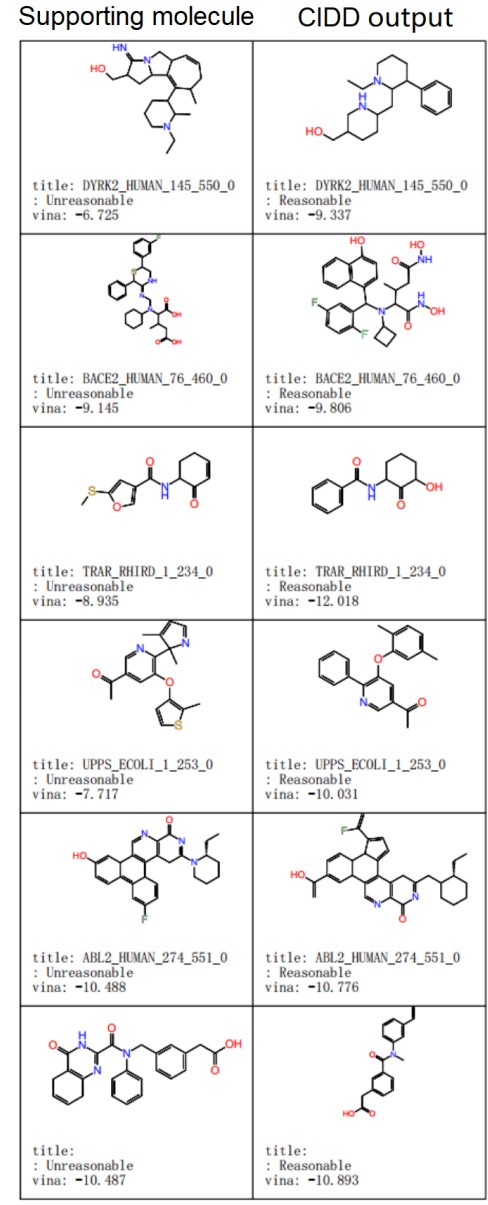

| Supporting molecule | CIDD output | Supporting molecule | CIDD output |
|---|---|---|---|

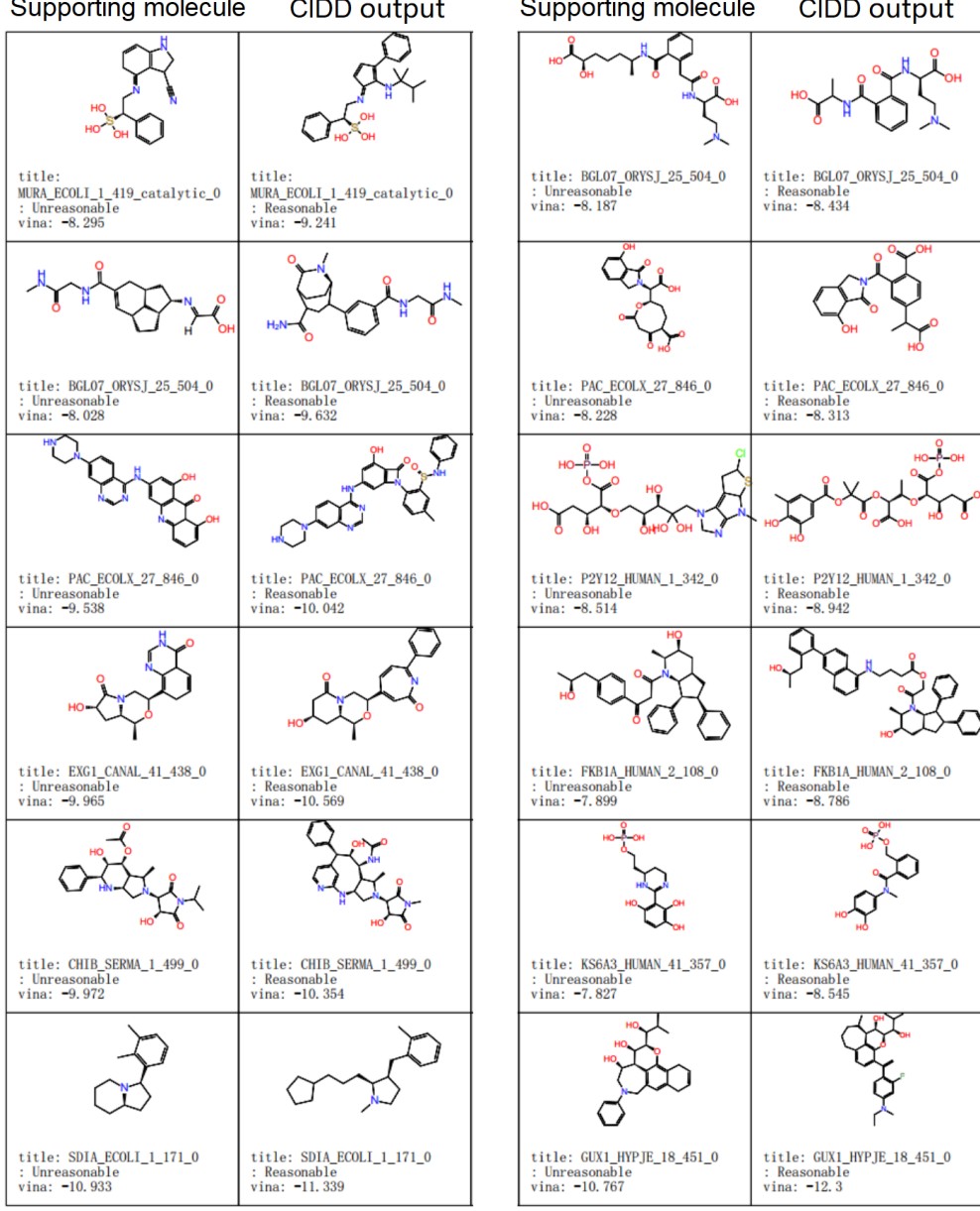

| | | | |
|---|---|---|---|
| title: MURA_ECOLI_1_419_catalytic_0 : Unreasonable vina: -8.295 | title: MURA_ECOLI_1_419_catalytic_0 : Reasonable vina: -9.241 | title: BGL07_ORYSJ_25_504_0 : Unreasonable vina: -8.187 | title: BGL07_ORYSJ_25_504_0 : Reasonable vina: -8.434 |
| title: BGL07_ORYSJ_25_504_0 : Unreasonable vina: -8.028 | title: BGL07_ORYSJ_25_504_0 : Reasonable vina: -9.632 | title: PAC_ECOLX_27_846_0 : Unreasonable vina: -8.228 | title: PAC_ECOLX_27_846_0 : Reasonable vina: -8.313 |
| title: PAC_ECOLX_27_846_0 : Unreasonable vina: -9.538 | title: PAC_ECOLX_27_846_0 : Reasonable vina: -10.042 | title: P2Y12_HUMAN_1_342_0 : Unreasonable vina: -8.514 | title: P2Y12_HUMAN_1_342_0 : Reasonable vina: -8.942 |
| title: EXG1_CANAL_41_438_0 : Unreasonable vina: -9.965 | title: EXG1_CANAL_41_438_0 : Reasonable vina: -10.569 | title: FKB1A_HUMAN_2_108_0 : Unreasonable vina: -7.899 | title: FKB1A_HUMAN_2_108_0 : Reasonable vina: -8.786 |
| title: CHIB_SERMA_1_499_0 : Unreasonable vina: -9.972 | title: CHIB_SERMA_1_499_0 : Reasonable vina: -10.354 | title: KS6A3_HUMAN_41_357_0 : Unreasonable vina: -7.827 | title: KS6A3_HUMAN_41_357_0 : Reasonable vina: -8.545 |
| title: SDIA_ECOLI_1_171_0 : Unreasonable vina: -10.933 | title: SDIA_ECOLI_1_171_0 : Reasonable vina: -11.339 | title: GUX1_HYPJE_18_451_0 : Unreasonable vina: -10.767 | title: GUX1_HYPJE_18_451_0 : Reasonable vina: -12.3 |

| Supporting molecule | CIDD output | Supporting molecule | CIDD output |
|---|---|---|---|

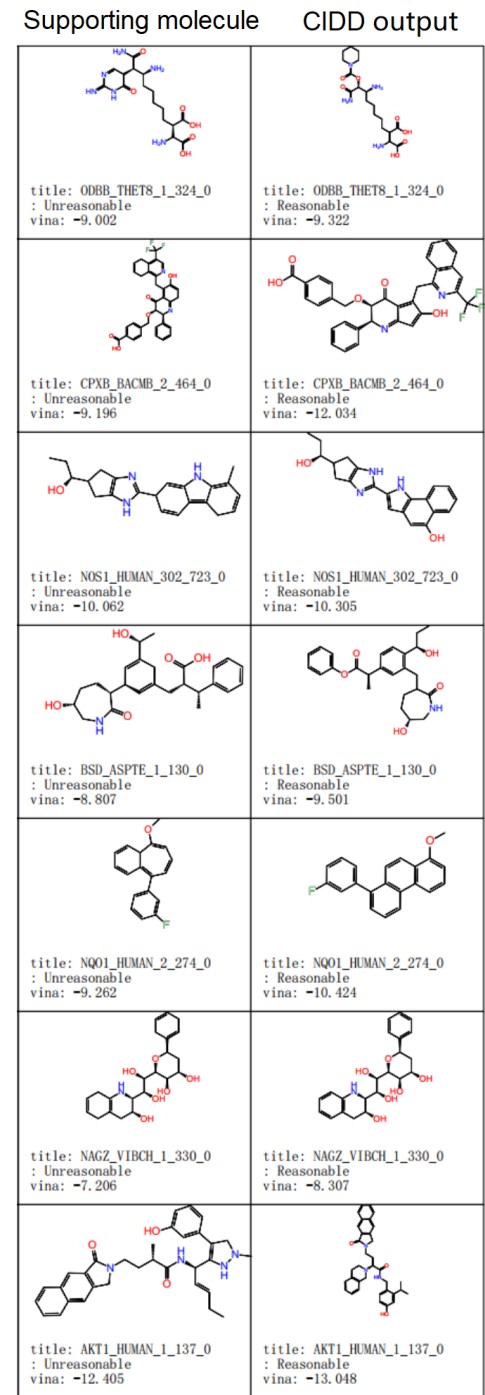

| | | | |
|---|---|---|---|
| title: HDHA_ECOLI_1_255_0 : Unreasonable vina: -8.511 | title: HDHA_ECOLI_1_255_0 : Reasonable vina: -9.995 | title: ODBB_THET8_1_324_0 : Unreasonable vina: -9.002 | title: ODBB_THET8_1_324_0 : Reasonable vina: -9.322 |
| title: IDHP_HUMAN_40_452_0 : Unreasonable vina: -8.039 | title: IDHP_HUMAN_40_452_0 : Reasonable vina: -9.819 | title: CPXB_BACMB_2_464_0 : Unreasonable vina: -9.196 | title: CPXB_BACMB_2_464_0 : Reasonable vina: -12.034 |
| title: NOS1_HUMAN_302_723_0 : Unreasonable vina: -9.495 | title: NOS1_HUMAN_302_723_0 : Reasonable vina: -9.579 | title: NOS1_HUMAN_302_723_0 : Unreasonable vina: -10.062 | title: NOS1_HUMAN_302_723_0 : Reasonable vina: -10.305 |
| title: CAT_ECOLX_1_219_0 : Unreasonable vina: -9.2 | title: CAT_ECOLX_1_219_0 : Reasonable vina: -13.057 | title: BSD_ASPTE_1_130_0 : Unreasonable vina: -8.807 | title: BSD_ASPTE_1_130_0 : Reasonable vina: -9.501 |
| title: ABL2_HUMAN_274_551_0 : Unreasonable vina: -10.702 | title: ABL2_HUMAN_274_551_0 : Reasonable vina: -11.583 | title: NQO1_HUMAN_2_274_0 : Unreasonable vina: -9.262 | title: NQO1_HUMAN_2_274_0 : Reasonable vina: -10.424 |
| title: BTRN_BACCI_2_250_0 : Unreasonable vina: -8.786 | title: BTRN_BACCI_2_250_0 : Reasonable vina: -8.951 | title: NAGZ_VIBCH_1_330_0 : Unreasonable vina: -7.206 | title: NAGZ_VIBCH_1_330_0 : Reasonable vina: -8.307 |
| title: AKT1_HUMAN_1_137_0 : Unreasonable vina: -10.1 | title: AKT1_HUMAN_1_137_0 : Reasonable vina: -11.478 | title: AKT1_HUMAN_1_137_0 : Unreasonable vina: -12.405 | title: AKT1_HUMAN_1_137_0 : Reasonable vina: -13.048 |

Supporting molecule    CIDD output        Supporting molecule    CIDD output

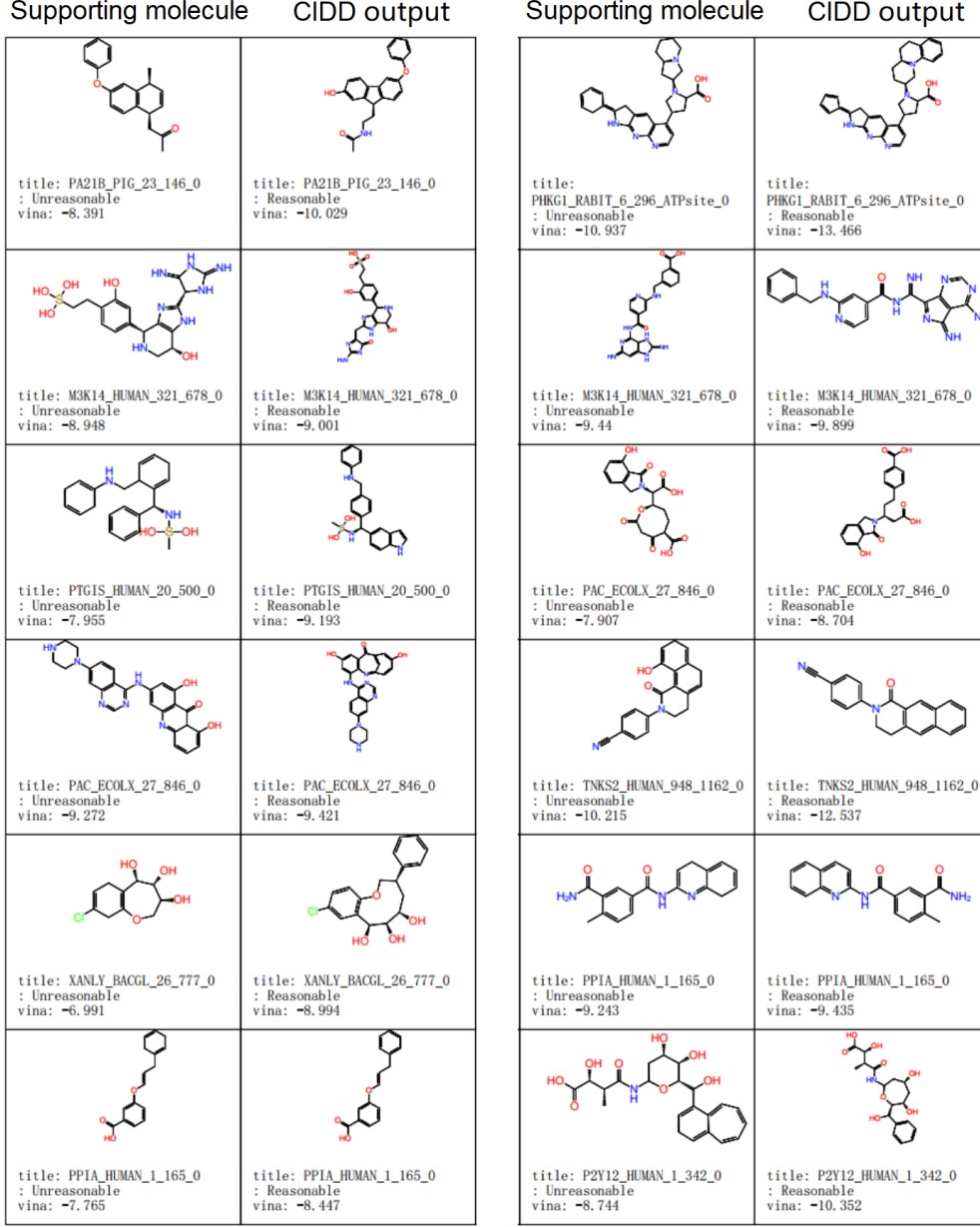

| Supporting molecule | CIDD output | Supporting molecule | CIDD output |
|---|---|---|---|

| title: P2Y12_HUMAN_1_342_0
: Unreasonable
vina: -8.31 | title: P2Y12_HUMAN_1_342_0
: Reasonable
vina: -9.509 | title: EXG1_CANAL_41_438_0
: Unreasonable
vina: -10.072 | title: EXG1_CANAL_41_438_0
: Reasonable
vina: -10.17 |
| title: MENE_BACSU_2_486_0
: Unreasonable
vina: -8.029 | title: MENE_BACSU_2_486_0
: Reasonable
vina: -8.629 | title: SQHC_ALIAD_1_631_0
: Unreasonable
vina: -9.519 | title: SQHC_ALIAD_1_631_0
: Reasonable
vina: -12.291 |
| title: CHIB_SERMA_1_499_0
: Unreasonable
vina: -7.389 | title: CHIB_SERMA_1_499_0
: Reasonable
vina: -9.388 | title: SIR3_HUMAN_117_398_0
: Unreasonable
vina: -10.258 | title: SIR3_HUMAN_117_398_0
: Reasonable
vina: -10.419 |
| title: DYRK2_HUMAN_145_550_0
: Unreasonable
vina: -7.898 | title: DYRK2_HUMAN_145_550_0
: Reasonable
vina: -9.098 | title: NOS3_HUMAN_65_480_0
: Unreasonable
vina: -9.821 | title: NOS3_HUMAN_65_480_0
: Reasonable
vina: -10.535 |
| title: NR1H4_HUMAN_258_486_0
: Unreasonable
vina: -8.974 | title: NR1H4_HUMAN_258_486_0
: Reasonable
vina: -9.969 | title: BACE2_HUMAN_76_460_0
: Unreasonable
vina: -9.302 | title: BACE2_HUMAN_76_460_0
: Reasonable
vina: -9.947 |
| title: BACE2_HUMAN_76_460_0
: Unreasonable
vina: -9.473 | title: BACE2_HUMAN_76_460_0
: Reasonable
vina: -9.828 | title: SDIA_ECOLI_1_171_0
: Unreasonable
vina: -8.889 | title: SDIA_ECOLI_1_171_0
: Reasonable
vina: -13.779 |
| title: TRAR_RHIRD_1_234_0
: Unreasonable
vina: -10.182 | title: TRAR_RHIRD_1_234_0
: Reasonable
vina: -13.112 | title: TRAR_RHIRD_1_234_0
: Unreasonable
vina: -9.69 | title: TRAR_RHIRD_1_234_0
: Reasonable
vina: -15.856 |

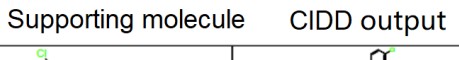

| Supporting molecule | CIDD output | Supporting molecule | CIDD output |
|---|---|---|---|

title: POL_FOAMV_861_1060_0
: Unreasonable
vina: −7.797

title: POL_FOAMV_861_1060_0
: Reasonable
vina: −8.475

title: HDHA_ECOLI_1_255_0
: Unreasonable
vina: −8.698

title: HDHA_ECOLI_1_255_0
: Reasonable
vina: −9.535

# I  Limitation on Computational Cost and Scalability

We acknowledge that our framework introduces additional computational steps due to the use of LLM-based modules. Below, we provide a detailed discussion of the computational overhead, empirical runtime, and justification for its practicality in the context of drug discovery.

**Computational Overhead of LLM Pipelines.**  The runtime and token-related costs associated with repeated LLM invocations—such as scaffold analysis, molecule generation, self-reflection, and candidate selection—can introduce non-trivial computational overhead. However, several factors mitigate this concern:

**LLM Inference Efficiency Is Rapidly Improving.**  Recent advances in model quantization, prompt batching, and optimized inference are dramatically reducing the cost of multi-step reasoning. For example, DeepSeek-V3/R1 achieves approximately 2.19M output tokens per dollar, reflecting a 60–70% cost reduction compared to GPT-4 in 2023. Frameworks such as vLLM [18], vTensor [36], and Splitwiser [1] further improve efficiency: vLLM enables up to $24\times$ throughput; vTensor achieves $2\times$ speedups with 70% less GPU memory; and Splitwiser reduces latency by $1.7\times$ through prompt colocation. Hardware accelerators like TensorRT-LLM also support highly scalable domain-specific pipelines. These trends suggest that the cost of multi-step reasoning is unlikely to remain a long-term bottleneck.

**Empirical Runtime of Our Pipeline.**  To provide a more quantitative view, we report the empirical runtime of our full pipeline. Based on our experiments with the MolCraft model (used as the SBIG module in our method), under the setting of 100 sampling steps, it takes approximately 300 seconds to generate 100 molecules. For competitive baselines, previous work reports that the autoregressive method Pocket2Mol requires around 2827 seconds to generate the same number of molecules, and diffusion-based methods such as TargetDiff and DecompDiff require approximately 3428 seconds and 6189 seconds, respectively.

As our method builds upon base SBDD models as the SBIG module, we only consider the additional time introduced by the LEDD module. With parallel LLM API requests, the total time required for generating 100 molecules is approximately 15 minutes in our experiments. Importantly, this added cost does not change the order of magnitude of the overall runtime.

The majority of this overhead stems from LLM inference latency, which is common across recent works that utilize LLMs. Nevertheless, this is unlikely to remain a bottleneck as inference efficiency continues to improve. For instance, recent advances in Diffusion-LLMs [31] have demonstrated inference speeds exceeding 2000 tokens per second, suggesting the potential for up to a $20\times$ acceleration of our pipeline in the near future.

**Molecule Design Is Not a Real-Time Task.**  Unlike latency-sensitive domains such as recommendation systems or autonomous control, molecule design prioritizes quality over speed. Drug discovery workflows are inherently iterative and require careful validation. While LLM-agent pipelines incur additional compute, their ability to generate higher-quality candidates with stronger binding plausibility aligns with the broader goal of improving experimental success rates, rather than immediate response times.

**Relative Cost Compared to Wet-Lab Validation.** In the broader context of drug development—which routinely spans several years and billions of dollars—LLM-related inference costs are marginal. The true bottlenecks lie in experimental synthesis, biochemical testing, and preclinical validation. Even modest improvements in in silico candidate quality can translate to significant downstream savings. Thus, the human-effort and resource reductions enabled by automated multi-step reasoning justify the modest computational overhead.

In summary, while our method introduces additional computational cost, it remains within a practical and acceptable range and does not diminish the significance, effectiveness, or applicability of our contributions. These tradeoffs will continue to improve as inference efficiency advances.

## J    Results on more targets

we tested CIDD on additional targets: 1sa1, 1t8i and 7rpz, all known for their structural complexity and variability in ligand scaffolds. For 1sa1, Tubulin presents a challenging drug target because its binding sites are highly dynamic, involving extensive protein–protein interfaces; furthermore, the ligand in this complex, podophyllotoxin, is a complex natural product scaffold[28]. For 1t8i, Human DNA Topoisomerase I presents a challenging drug target because its catalytic mechanism involves a highly dynamic binding site that accommodates DNA; moreover, the ligand in this initial complex is camptothecin, a natural product scaffold rather than a traditionally rationally designed small molecule[32]. For 7rpz, KRASG12D is difficult to inhibit due to its high GDP/GTP affinity and absence of a nucleophilic residue near the switch II pocket, which hampers covalent strategies and limits prior chemotypes. The ligand uniquely overcomes these barriers with a pyrido[4,3-d]pyrimidine scaffold, fully engages the pocket, and achieves picomolar noncovalent inhibition with strong in vivo efficacy[35].

Table 5: Comparison between MolCRAFT and CIDD on different targets

| Target | Method | Success Ratio ↑ | Vina Score ↓ | QED ↑ | SA ↑ | MRR ↑ |
|--------|--------|-----------------|--------------|-------|------|-------|
| 1sa1 | MolCRAFT | 27.78% | -9.78 | 0.56 | 0.66 | 35.56% |
| 1sa1 | CIDD | 54.44% | -10.47 | 0.60 | 0.71 | 62.22% |
| 1t8i | MolCRAFT | 9.64% | -7.96 | 0.37 | 0.57 | 60.24% |
| 1t8i | CIDD | 27.71% | -8.38 | 0.39 | 0.62 | 68.67% |
| 7rpz | MolCRAFT | 1.35% | -9.71 | 0.27 | 0.57 | 1.35% |
| 7rpz | CIDD | 14.86% | -10.35 | 0.32 | 0.61 | 14.86% |

## K    Ablation Study: Reasoning Design vs. Simple Prompting

To investigate the source of our framework's effectiveness, we conducted an ablation study comparing different prompting strategies. Specifically, we replaced our original chain-of-thought (CoT), domain-informed prompt with a simplified instruction: *"Based on the pocket and interaction analysis, modify the original molecule."*

This experiment isolates the effect of prompt engineering by using the same underlying LLM and tools, but with varying reasoning structure.

Table 6: Ablation results comparing the impact of prompt complexity on model performance.

| Method | SA ↑ | MRR ↑ | Success ↑ |
|--------|------|-------|-----------|
| MolCRAFT (baseline) | 0.685 | 58.47% | 13.72% |
| CIDD (simple prompt) | 0.678 | 63.69% | 19.76% |
| CIDD (CoT prompt, ours) | 0.735 | 81.74% | 34.59% |

Despite using the same LLM and external tools, the simplified prompting strategy performs significantly worse than our CoT-guided approach, and only marginally better than the MolCRAFT

baseline. This highlights that the primary gains of our method stem from structured, domain-informed reasoning embedded in the prompt design, rather than from tool integration alone.

## L   Extension: Applicability to Hit-to-Lead Optimization

While the CIDD framework was not originally designed for hit-to-lead optimization, it can be readily adapted to such scenarios. To explore this possibility, we initialized the generation process—and the input to the LEDD module—using ground-truth ligands from the CrossDocked test set.

Despite these ligands already demonstrating strong synthetic accessibility and structural quality, CIDD was able to generate new molecules that exhibited further improvements in potency (as measured by docking score), synthesizability (QED and SA), and overall reasonability (MRR and success ratio).

Table 7: Evaluation of hit-to-lead potential by initializing CIDD with known binders from the CrossDocked test set.

| Method | Vina Score ↓ | QED ↑ | SA ↑ | MRR ↑ | Success Ratio ↑ |
|---|---|---|---|---|---|
| Original Ligands | -7.53 | 0.473 | 0.738 | 0.8272 | 22.22% |
| CIDD (generated) | -8.22 | 0.528 | 0.752 | 0.9136 | 37.04% |

These results demonstrate that, even when starting from already high-quality ligands, our framework is capable of discovering improved candidates through domain-aware multi-agent reasoning.

We note that fully supporting hit-to-lead tasks would require adapting the pipeline structure, modifying the chain-of-thought prompts, and refining the integration of design tools to better align with lead optimization objectives. Nonetheless, the observed performance underscores the broader applicability of our framework and the potential of structured LLM reasoning in more advanced drug development stages.

We plan to incorporate support for hit-to-lead optimization more formally in future work.

## M   Effect of Re-Docking SBIG-Generated Molecules

To ensure molecular stability and structural correctness, our pipeline includes a re-docking step following initial molecule generation. The rationale behind this design choice is that conformations directly produced by generative models may suffer from issues such as atomic clashes or physically unstable geometries, which can negatively affect downstream interaction analysis and overall pipeline reliability. This concern has also been discussed in recent work [? ]. Re-docking provides a more stable and physically reasonable starting point for the LEDD module.

To evaluate whether this re-docking step is necessary, we conducted an ablation study comparing the performance of our pipeline with and without re-docking. Specifically, we initialized the LEDD module with either (1) re-docked poses or (2) the original generated conformations. All experiments used DeepSeek-V3 as the LLM.

Table 8: Ablation study evaluating the effect of re-docking versus directly using generated conformations.

| Method | Vina Score ↓ | QED ↑ | SA ↑ | MRR ↑ | Success Ratio ↑ |
|---|---|---|---|---|---|
| Re-docking | -8.49 | 0.565 | 0.714 | 76.00% | 29.66% |
| Original conformation | -8.403 | 0.575 | 0.721 | 76.09% | 28.03% |

As shown in the table, both variants achieve comparable results, with re-docking offering slight improvements in docking score and success ratio, while direct usage of the original conformation shows marginally better QED and SA. Overall, this suggests that our framework is robust to this design decision.

While re-docking provides additional geometric stability, the small performance gap indicates that either choice is acceptable in practice, depending on computational tradeoffs. This flexibility may be useful in settings where runtime is more constrained. .

