# OpenReview forum: "CIDD: Collaborative Intelligence for Structure-Based Drug Design Empowered by LLMs"
_NeurIPS.cc/2025/Conference — NeurIPS 2025 poster_

### Official Review · Reviewer_DBAy · 2025-06-17

**Clarity:** 2
**Significance:** 1
**Originality:** 1
**Rating:** 3
**Confidence:** 4

**Summary:**

Through this paper, the authors introduce the Collaborative Intelligence Drug Design (CIDD) framework. The CIDD framework employs CoT reasoning of LLMs to generate 3D molecules that are compatible with protein pockets and exhibit favorable chemical properties. The authors also propose a new metric, the Molecule Reasonable Ratio (MRR), to measure structural rationality of generated molecules.

**Questions:**

- The proposed CIDD framework heavily relies on the backbone LLM model, and is therefore inevitably vulnerable to hallucination or reasoning errors. Are there potential ways to overcome or alleviate this issue?
- Please see *Weaknesses* above for my main concerns about this paper.

**Ethical Concerns:**

["NO or VERY MINOR ethics concerns only"]

**Final Justification:**

I considered author rebuttal and read other reviewer's reviews. My concerns on the novelty and other limitations of the proposed method were not fully resolved during the rebuttal, therefore I keep my original score.

**Limitations:**

The authors did not discuss the potential negative societal impact. The limitations of this work is very briefly discussed (in one sentence). However, there seem to be a lot of limitations that can possibly invoke interesting views to the readers. For example, the performance of the CIDD framework heavily relies on the LLM model, and it can show degraded performance with less powerful or open-source LLMs. Moreover, how to overcome or alleviate the hallucination of the backbone LLM can be an interesting discussion.

**Paper Formatting Concerns:**

--

**Quality:**

2

**Strengths And Weaknesses:**

**Strengths:**
- The writing and figures are clear and overall easy to follow.
- The direction of integrating SBDD with LLMs is interesting.
- The proposed CIDD framework is modular and interpretable.

**Weaknesses:**
- The authors did not provide the codebase to reproduce the results.
- The main weakness of this paper is its weak novelty. The framework proposed in this paper relies on many external rule-based libraries. Rather than generating molecules through the collaboration of SBDD model and LLM, it seems to be a heuristic framework through the combination of several rules such as BRICS, external chemical analysis tools, domain-knowledge prompts, etc. For example, the paper uses multiple modules for the LLM-enhanced drug designer (LEDD) step, and there is no in-depth ablation study for each module.
- In addition, the use of external experts or oracles is not a fair comparison with existing SBDD models. In the field of drug discovery, sample efficiency, i.e., oracle call budget, is very important [1], and CIDD uses more oracles such as the external interaction analysis tool than existing models.
- Furthermore, the computational cost of the proposed method can be high due to the repeated cycles with many external modules and multiple LLM inference steps. However, there is no discussion or comparison with existing SBDD methods regarding computational efficiency.
- The justification for the new metric, MRR, seems to be insufficient. The results of how MRR is measured for known drugs are required. Furthermore, as also mentioned in the paper, MRR uses heuristic rules, and it is not explained how these rules are supported by existing chemical knowledge. For example, in Algorithm 1, why are carbonyl and imine groups excluded in calculating MRR? A detailed explanation should be included.

---
**References:**

[1] Gao et al., Sample Efficiency Matters: A Benchmark for Practical Molecular Optimization, NeurIPS Track on Datasets and Benchmarks, 2022.

---

> ### Author Rebuttal · Authors · 2025-07-30
>
> We sincerely thank the reviewer for their thoughtful feedback. Below are our responses to the main points:
>
> ## Regarding Code
> We apologize for not including the code in the initial submission. Due to file-upload restrictions during the rebuttal phase, we cannot share it publicly now but are fully committed to open-sourcing upon acceptance. In LLM-based systems like ours, prompt design is central to model behavior. To support reproducibility, **we provide detailed prompt templates and example outputs in Appendix**, and step-by-step pipeline descriptions (main text). **If any part of the prompt design is unclear, we’re happy to clarify during the rebuttal.**
>
> ## Regarding Novelty and Contribution
> We thank the reviewer for their thoughtful comments. Our contribution may have been misunderstood as a heuristic combination of tools, but in fact, **the LLM plays a central and decisive role** in our framework, reasoning over structured molecular knowledge and coordinating all key design decisions. Rather than a simple toolchain, our method is definitely a novel **collaboration between LLMs and 3D SBDD models**, where the LLM drives the refinement process with domain-informed reasoning.
>
> ### Integrating LLMs into SBDD is Non-Trivial and Scientifically Novel
> LLMs are not inherently designed to understand molecular graphs, 3D structures, or protein-ligand interactions. Their reasoning is based on symbolic text, not geometry or chemistry. Therefore, applying LLMs to tasks that require spatial reasoning and  chemistry knowledge—as is essential in SBDD—is a major challenge. Our contribution lies in bridging this gap through a carefully designed pipeline.
>
> ### Ablation Study: Effectiveness Comes from LLM Reasoning Design, not external rule-based libraries.
>
> Following the reviewer’s suggestion, we conducted an ablation study to further support the non-triviality and novelty of our LLM-SBDD integration. Specifically, we replaced our original chain-of-thought, domain-informed prompts with a simple instruction: “Based on the pocket and interaction analysis, modify the original molecule.”
>
> | Method| SA ↑ | MRR ↑| Success ↑ |
> |-------------|--------|---------|----------|
> | MolCRAFT|0.685|58.47%|13.72%|
> | CIDD (simple prompt)|0.678|63.69%|19.76%|
> | CIDD (CoT prompt)|0.735 |81.74%| 34.59%|
>
> Notably, despite using the same LLM and tools, the simplified prompting strategy performs significantly worse than the CoT-guided version and is only marginally better than the MolCRAFT baseline. **This underscores that the effectiveness of our framework lies not in tool usage, but in domain-informed, structured reasoning via prompt design.**
>
> **We are prepared to conduct additional ablation studies should reviewers request further clarification or analysis.**
>
> ### The Core Novelty: A Domain-Guided, LLM-Centered Framework
>
> Our key innovation is a framework where the LLM serves not just as a molecule generator, but as a decision-making coordinator that mimics expert reasoning in structure-based drug design (SBDD). We encode domain knowledge into prompts, allowing the LLM to interpret chemical and structural constraints in a human-like, structured way. Specialized tools (e.g., PLIP, RDKit) handle atomic-level tasks like interaction extraction or fragment generation, while the LLM focuses on high-level reasoning and decision-making. This division of labor plays to each component’s strength and offsets the LLM’s limitations with raw 3D geometry.
>
> This mirrors how medicinal chemists operate—translating 3D interactions into symbolic reasoning before modifying structures. Our framework replicates this workflow, refining SBDD outputs into drug-like molecules without relying on oracle-based optimization.
>
> ## Regarding oracle budget
>
> We thank the reviewer for raising this concern and offer the following clarifications:
>
> ### CIDD is not an oracle-based optimization framework.
> Our goal is to preserve the 3D interaction patterns from the initial SBDD output while improving drug-likeness (e.g., stability, synthesizability, plausibility). We do not query any drug-likeness-related oracles (e.g., QikProp, MRR, SA, QED) during generation; these metrics are used only post hoc for evaluation. This clearly distinguishes our method from optimization pipelines that rely on repeated oracle feedback.
>
> ### PLIP enables interaction representation—not optimization.
> Since LLMs cannot directly process 3D structures, we use PLIP to extract symbolic interaction patterns (e.g., hydrogen bonds, hydrophobic contacts) from the SBDD-generated molecule. These are passed to the LLM in natural language to support reasoning. PLIP is used solely as a structural-to-symbolic translator—not a scoring or feedback oracle.
>
> ### Relations with existing SBDD model
> As we are not oracle based optimization method, we believe it is still a fair comparasion with existing SBDD model. And actually we are not "competitive" with existing SBDD models. Instead, we are trying to build upon, or to say help them to solve unresolved problems such as structure reasonability and syntehzi ability.
>
> ## Regarding computational costs
> We acknowledge the reviewer’s concern regarding the runtime and token-related costs associated with repeated LLM invocations (e.g., scaffold analysis, molecule design, reflection, and selection). Indeed, LLM-agent pipelines—like ours, and as seen in many other LLM-agent use cases—can incur non-trivial computational overhead. However, we would like to offer the following justifications:
>
> ### Cost Is Becoming a Lesser Barrier
> The cost of multi-step LLM reasoning is rapidly shrinking thanks to advances in quantization, batching, and optimized inference. DeepSeek-V3/R1 achieves ~$0.55M/input and ~$2.19M/output tokens—a 60–70% drop from GPT-4’s 2023 cost. Frameworks like vLLM[1], vTensor[2], and Splitwiser[3] further boost efficiency: vLLM offers up to 24× throughput, vTensor gives 2× speedups with 70% less GPU memory, and Splitwiser reduces latency by 1.7× via prompt-generation colocation. With accelerators like TensorRT-LLM, domain-aware multi-step pipelines are becoming increasingly scalable. Overall, **computational overhead is shrinking** and is unlikely to be a major limitation in the future.
>
> [1] Kwon et al., SOSP 2023 — PagedAttention for efficient memory management in LLM serving.
>
> [2] Xu et al., CoRR 2024 — vTensor: virtual tensor management for efficient LLM inference.
>
> [3] Aali et al., arXiv 2025 — Splitwiser: latency-optimized LLM inference under resource constraints.
>
> ### Traditional Drug Development Is Far More Costly
>
> Drug development is slow and costly, often taking years and billions of dollars to bring a single drug to market. The true bottleneck lies not in design time, but in the lengthy, expensive trial-and-error process of testing candidate molecules. Unlike real-time tasks like recommendation or autonomous driving, drug discovery prioritizes precision over speed. What matters is generating molecules with higher chances of experimental success—a goal our framework supports by improving the quality and viability of SBDD-generated candidates.
>
> Our framework does not replace traditional SBDD models but **accelerates their downstream refinement** through a structured, multi-agent pipeline that preserves key interactions while improving structural plausibility. **Although multi-step reasoning introduces some computational overhead, the significant reduction in human effort more than compensates—especially in large-scale or high-throughput scenarios.**
>
> We thank the reviewer for raising this point. **We will discuss this tradeoff in the Limitations section and note that ongoing advances in LLM efficiency continue to reduce these barriers.**
>
> ## Regarding new metric, MRR
>
> Defining strict criteria for drug-likeness remains elusive, yet long-standing empirical evidence has delineated recurrent structural motifs. In cyclic scaffolds, extranuclear electron distributions can enforce stability when specific electronic configurations, notably continuous sp² hybridization, are satisfied[1, 2]. Carbonyl and imine substituents often conform to sp² hybridization yet fail to establish a fully delocalized π-system around the ring—a feature frequently observed in approved drugs[3, 4, 5]. The MRR formalizes these principles, accurately distinguishing drug-like from non-drug-like molecules: **85.90% of FDA-approved small molecules comply with MRR criteria**, substantially exceeding that of current 3D-SBDD models. By focusing on ring systems, present in **83.31% of approved drugs**, MRR captures a fundamental chemical constraint: stable rings adopt either fully saturated or fully aromatic configurations, thereby conferring thermodynamic stability, synthetic tractability, and favorable interaction profiles—properties systematically exploited in both natural products and rationally designed therapeutics.
>
> [1] Morsch et al., Organic Chemistry, LibreTexts, Ch. 15.2.
>
> [2] Aromatic Compounds, ChemistryTalk, accessed 30 Jul 2025.
>
> [3] Saudagar PS et al., Biotechnol Adv, 26(4):335–351, 2008.
>
> [4] Poupaert JH et al., J Med Chem, 27(1):76–78, 1984.
>
> [5] Stezowski JJ, J Am Chem Soc, 98(19):6012–6018, 1976.
>
> ## Regarding hallucination or reasoning errors.
>
> We thank the reviewer for raising this important concern. We mitigate hallucination and reasoning errors via:
>
> Controlled decoding (low temperature) to reduce randomness and improve consistency.
>
> Instructional prompting with clear reasoning steps and domain constraints, leveraging strong instruction-following models (GPT-4o, DeepSeek).
>
> Self-correction, where invalid molecules trigger reflection and regeneration, enhancing robustness without external feedback.
>
> We will include this discussion and future directions in the final version.
>
> ## Regarding Limitation
> We thank the reviewer for raising these important concerns. We will include discussions on the societal impact, as well as the  limitations mentioned above, in the final version.

---

> > ### Comment · Reviewer_DBAy · 2025-08-07
> >
> > I appreciate the authors for trying to clarify some of the concerns I raised. However, I am not fully convinced of the novelty and other limitations of the proposed method. Regardless of PLIP is used as a scoring oracle or not, its call can be expensive and should be included in the budget. Regarding the computational costs, could the authors provide a direct comparison of wall-clock time? This would be helpful for readers' understanding.

---

> > > ### Author Response · Authors · 2025-08-07
> > >
> > > Dear reviewer,
> > >
> > > We sincerely appreciate your continued feedback and the opportunity to further clarify.
> > >
> > > ## Computational Cost of PLIP
> > >
> > > Regarding the computational cost of PLIP, to address this directly, we conducted an evaluation and found that PLIP takes approximately **22 seconds to process 100 protein–ligand complexes, which translates to around 0.22 seconds per complex with one cpu core.**
> > >
> > > Given this, we believe the overhead introduced by PLIP is relatively modest in practice.
> > >
> > > ## Novelty and Contributions
> > > As for the novelty, while we truly understand and respect your perspective, we would like to offer a bit more context in the hope that it might help better convey our intended contribution.
> > >
> > > 1. **From a conceptual perspective**, this work represents a **pioneering effort** in applying large language models to structure-based drug design. We are **the first** to formally identify and attempt to solve the structure reasonability problem in 3D SBDD, which has not been explicitly addressed in prior research. Most existing studies focus on optimizing specific, quantifiable metrics such as docking scores, QED, or synthetic accessibility. While these metrics are important, they **fail to fully capture** the structural plausibility and chemical soundness **required in real-world drug discovery**. To address this gap, we introduce **new metrics** to characterize the problem, and we propose a **novel framework** that integrates LLMs with 3D SBDD models to tackle it. Altogether, we believe our work offers a new perspective to the field, opens promising directions for incorporating LLMs into SBDD, and addresses an important yet previously overlooked challenge.
> > >
> > > 2. **From a technical perspective**, our justifications and ablation studies demonstrate that applying LLMs to SBDD tasks is **far from trivial**. **Our ablation** has shown that the use of knowledge-guided prompts with Chain-of-Thought reasoning are **essential** in enabling the framework to outperform conventional SBDD models in molecule quality. This highlights the critical role of LLMs in our approach and supports our contribution in designing prompts that allow LLMs to better leverage their own strengths and compensate for their weaknesses through our tailored collaborative pipeline.
> > >
> > > In summary, we believe our paper introduces a novel framework with meaningful technical contributions, while identifying and addressing an important gap in current research.
> > >
> > > We also acknowledge that our approach has limitations, and we are truly grateful to the reviewers for pointing them out. We will carefully discuss these limitations in the revised manuscript. Nevertheless, we believe our work makes a important contribution to the community, not only by identifying and addressing a previously neglected problem, but also by offering a new perspective that may inspire future research toward more effective and reliable drug design pipelines.
> > >
> > >
> > > Once again, we sincerely thank the reviewer for their time, thoughtful comments, and constructive suggestions. We hope these additional clarifications help address your concerns and demonstrate the value of our contribution. Your feedback is highly appreciated and has been important in helping us improve this work. We would be truly grateful if our responses have moved the evaluation closer to a positive outcome.

---

> > > ### Author Response · Authors · 2025-08-08
> > >
> > > We would like to sincerely thank the reviewer once again for their time, effort, and thoughtful feedback on our work. We hope our rebuttal has adequately addressed the concerns raised. If there are any further questions or points that require discussion or clarification, please do not hesitate to let us know.
> > >
> > > **If possible, we would greatly appreciate it if the reviewer could kindly let us know whether this may lead to a change in the score, and, if so, what the updated score might be.**
> > >
> > > Thank you again for your time and consideration.

---

> > > > ### Comment · Reviewer_DBAy · 2025-08-08
> > > >
> > > > Thank you for your reply. However, what I meant by “wall clock time” was the time it takes to run the proposed framework (for example, to generate 100 molecules). And in order to properly compare efficiency, I think a comparison with the baseline is necessary.

---

> > > > > ### Author Response · Authors · 2025-08-08
> > > > >
> > > > > Dear reviewer,
> > > > >
> > > > > Thanks for the response and clarification. We will provide the time for the full pipeline.
> > > > >
> > > > > Based on our own tests of MolCraft (used as the SBIG module in our method), under the setting of 100 sampling steps, it takes approximately 300 seconds to generate 100 molecules. For competitive baselines, previous work reports that the autoregressive method Pocket2Mol requires around 2827 seconds to generate the same number of molecules, and diffusion-based methods such as TargetDiff and DecompDiff require approximately 3428 seconds and 6189 seconds, respectively.
> > > > >
> > > > > As our method builds upon base SBDD models as our SBIG module, we only consider the additional time introduced by the LEDD module. With parellel LLM api request, the total time required for generating 100 molecules is approximately 15 minutes in our experiments. Importantly, this added cost does not change the order of magnitude of the overall runtime.
> > > > >
> > > > > The majority of this overhead stems from LLM inference latency, which is common across recent works that utilize LLMs. Nevertheless, we believe this will not remain a bottleneck as inference efficiency continues to improve. For example, recent advances in **Diffusion-LLMs[1]** have demonstrated inference speeds exceeding 2000 tokens per second, suggesting the potential for up to a **20× acceleration** of our pipeline in the near future.
> > > > >
> > > > > As we emphasized in our rebuttal, drug discovery is inherently expensive and time-consuming, where effectiveness and molecular quality are far more important than differences in generation speed. This is also why many existing structure-based drug design  methods, including the benchmark paper for this task, do not put computational cost in the main text and main result table, as it is not the primary focus.
> > > > >
> > > > > In summary, while our method introduces some additional computational overhead, it remains well within a practical and acceptable range, and does not diminish the significance, effectiveness, or applicability of our contributions.
> > > > >
> > > > > Thank you for pointing this out, which help refine our paper and will help provide readers with a clearer and more comprehensive understanding.
> > > > >
> > > > >
> > > > > [1] Song, Yuxuan, et al. "Seed Diffusion: A Large-Scale Diffusion Language Model with High-Speed Inference." arXiv preprint arXiv:2508.02193 (2025).

---

> > > > > ### Author Response · Authors · 2025-08-09
> > > > >
> > > > > We sincerely thank the reviewer once again for their valuable feedback, which has helped us refine and improve our paper.
> > > > >
> > > > > Since this was the only remaining question after our earlier exchanges and discussions, we take it as a hopeful indication that our previous clarifications were largely aligned with your expectations and that we have reached convergence on those points. For this final question, we respectfully hope that our responses and the supporting evidence provided will also be taken into consideration, and we truly appreciate your time and effort in evaluating our work.
> > > > >
> > > > > We are very grateful for the constructive suggestions you have provided, which will continue to guide us in further improving the clarity and quality of our paper.

---

### Official Review · Reviewer_M6aF · 2025-07-01

**Clarity:** 3
**Significance:** 3
**Originality:** 3
**Rating:** 5
**Confidence:** 3

**Summary:**

The authors tackle a central shortcoming of contemporary structure-based generative models: though 3D methods (e.g., diffusion or autoregressive approaches like MolCRAFT or Pocket2Mol) can propose molecules that fit tightly into protein pockets, they frequently yield chemically implausible scaffolds--strained rings, partial unsaturation or over-fused polycycles--that fall outside the realm of drug-like chemical space . To quantify this disconnect, they introduce the Molecular Reasonability Ratio (MRR), a rule-based metric that flags rings whose non-aromatic, non-saturated cores exhibit mixed sp^2/sp^3 hybridization (Cyclohexene- or diene-like motifs), revealing that existing 3D models achieve only ~38–59% MRR versus ~86% for FDA-approved drugs .

To bridge the gap between geometric complementarity and chemical viability, the paper presents CIDD, a two-stage, modular pipeline that couples:

1. SBIG, which uses any off-the-shelf 3D model to propose “raw” interaction-optimized scaffolds.
2. LEDD, which employs Chain-of-Thought prompts in a sequence of LLM-powered modules--Interaction Analysis, Design, Reflection and Selection--to iteratively refine those scaffolds into drug-like candidates .

On the CrossDocked2020 benchmark (100 test pockets × 10 molecules each), CIDD more than doubles the prior best combined success rate (binding + drug-likeness) from 15.7% to 34.6%, while boosting QED (≈0.50 to 0.58), SA score (≈0.68 to 0.74) and MRR (≈58–67% to 81.7%) without sacrificing docking affinity (mean Vina ≈–8.5) . These results demonstrate that integrating precise 3D interaction proposals with LLM-driven chemical reasoning can yield molecules that are both potent binders and structurally rational--paving the way for more interpretable, expert-inspired AI workflows in early-stage drug discovery.

**Questions:**

How well does CIDD generalize to diverse or challenging targets?

Can you provide detailed computational‐cost metrics?

What is the synthetic feasibility of your top candidates?

**Ethical Concerns:**

["NO or VERY MINOR ethics concerns only"]

**Final Justification:**

Authors did a good job on the rebuttal and solving my concerns. I believe this paper should be accepted.

**Limitations:**

Yes

**Quality:**

3

**Strengths And Weaknesses:**

## Strengths

By coupling any off-the-shelf 3D scaffold generator (SBIG) with an LLM-driven refinement loop, CIDD leverages both precise pocket fitting and human-like chemical intuition in a single pipeline.

The Molecular Reasonability Ratio (MRR) provides a clear, rule-based metric to expose and quantify chemically implausible scaffolds--an issue often hidden when optimizing purely for docking scores.

On the CrossDocked2020 benchmark, CIDD more than doubles the combined binding-and-drug-likeness success rate (from ~15.7 % to ~34.6 %) while also boosting QED, SA, and MRR scores without sacrificing docking affinity.

The clear SBIG→LEDD separation means practitioners can plug in newer 3D backbones or more advanced LLMs without retraining the entire system, facilitating future updates and improvements.

Chain-of-Thought prompts generate a transparent, step-by-step record—fragment analyses, flagged liabilities, planned edits, reflective assessments—that mirrors a medicinal chemist’s reasoning and supports human review.

## Weaknesses

Repeated LLM invocations per scaffold (analysis, design, reflection, selection) risk prohibitive runtimes and token-usage costs when screening large libraries, yet no wall-clock times, token counts or cost estimates are provided.

Validation is confined to a single in silico benchmark (CrossDocked2020) with relatively narrow chemical diversity. It remains unknown how CIDD performs on more challenging targets or proprietary datasets with different chemotype distributions.

While Chain-of-Thought prompting enhances interpretability, it may also introduce brittleness: small rephrasings or LLM updates could unpredictably alter design outcomes. The paper lacks any analysis of prompt sensitivity or strategies for automated prompt optimization.

---

> ### Author Rebuttal · Authors · 2025-07-30
>
> We sincerely thank the reviewer for their thoughtful evaluation and positive feedback on our paper. We appreciate the insightful comments and are happy to address the points raised below.
>
> ## Regarding costs
>
> We acknowledge the reviewer’s concern regarding the runtime and token-related costs associated with repeated LLM invocations (e.g., scaffold analysis, molecule design, reflection, and selection). Indeed, LLM-agent pipelines—like ours, and as seen in many other LLM-agent use cases—can incur non-trivial computational overhead. However, we would like to offer the following clarifications:
>
> ### Cost Is Becoming a Lesser Barrier
>
> The cost of multi-step LLM reasoning is rapidly declining thanks to advances like quantization, batching, and optimized inference engines. For example, DeepSeek-V3/R1 supports inference at $0.55M/input and $2.19M/output tokens—a 60–70% reduction from GPT-4’s 2023 pricing. Efficiency-focused frameworks such as vLLM[1], vTensor[2], and Splitwiser[3] further accelerate inference: vLLM achieves up to 24× higher throughput, vTensor offers 2× speedups with 70% less GPU memory, and Splitwiser reduces latency by 1.7× via co-located prompt and generation. Combined with hardware accelerators like TensorRT-LLM, these developments make domain-informed, multi-step LLM workflows increasingly practical and scalable. For example, with inference frameworks like vLLM, the DeepSeekV3 model can be deployed on 8×A100 80GB GPUs.
>
> In addition, attention-level optimizations have significantly accelerated LLM inference: FlashAttention‑2 achieves up to 4–5× speedup over standard attention [4], DuoAttention reduces memory usage by 2.55× and speeds up decoding by 2.18× [5], and MInference delivers up to 10× acceleration in pre-filling long contexts [6].
>
> These examples confirm that the computational and token overhead associated with multi-step LLM pipelines is **rapidly shrinking** and is unlikely to pose a significant obstacle in the foreseeable future.
>
> [1] Kwon, Woosuk, et al. "Efficient memory management for large language model serving with pagedattention." Proceedings of the 29th symposium on operating systems principles. 2023.
> [2] Xu, Jiale, et al. "vTensor: Flexible Virtual Tensor Management for Efficient LLM Serving." CoRR (2024).
> [3] Aali, A., Cardoza, A., & Capo, M. (2025). Splitwiser: Efficient LM inference with constrained resources. arXiv preprint arXiv:2505.03763.
> [4] Dao, T. FlashAttention‑2: Faster Attention with Better Parallelism and Work Partitioning. arXiv preprint arXiv:2307.08691.
> [5] Xiao, G. et al. DuoAttention: Efficient Long‑Context LLM Inference with Retrieval and Streaming Heads. In International Conference on Learning Representations (ICLR), 2025. arXiv preprint arXiv:2410.10819.
> [6] Jiang, H. et al. MInference 1.0: Accelerating Pre‑filling for Long‑Context LLMs via Dynamic Sparse Attention. In Conference on Neural Information Processing Systems (NeurIPS), 2024. arXiv preprint arXiv:2407.02490.
>
>
> ### Traditional Drug Development Is Far More Costly
>
> Drug development is slow and costly, often taking years and billions of dollars to bring a single drug to market. The true bottleneck lies not in design time, but in the lengthy, expensive trial-and-error process of testing candidate molecules. Unlike real-time tasks like recommendation or autonomous driving, drug discovery prioritizes precision over speed. What matters is generating molecules with higher chances of experimental success—a goal our framework supports by improving the quality and viability of SBDD-generated candidates.
>
> Our framework does not replace traditional SBDD models but **accelerates their downstream refinement** through a structured, multi-agent pipeline that preserves key interactions while improving structural plausibility. **Although multi-step reasoning introduces some computational overhead, the overall time savings from eliminating manual, expert-driven intervention far outweighs the additional inference cost**—especially in large-scale or high-throughput scenarios.
>
> ### Case Study: Runtime and Token Usage
> To provide a concrete estimate, we profiled a representative design iteration with the following statistics:
>
> Average runtime per design : ~1 minutes;  Total token usage per design: ~10000 tokens
>
> In this paper for each initial molecules we generate 5 designs sequentially so the cost would be 5 times. This can be adjusted based on requirements. In practice we found that one design is able to create molecules with better drug likeness and syteh
>
> Importantly, these operations are **highly parallelizable**, and large-scale molecules screening can be executed efficiently in batch across multiple machines, and still support generating large scale number of molecules in reasonable time.
>
> In summary, we thank the reviewer for raising this important point. We acknowledge that cost is a common concern for frameworks involving multiple LLM agents, and we will include a discussion of this limitation in the revised version of the paper. That said, we believe recent advances in model efficiency and inference infrastructure are rapidly reducing this barrier, as discussed above.
>
> ## Regarding more challenging targets
>
> We thank the reviewer for the helpful suggestion to evaluate CIDD on more challenging targets with diverse chemotype distributions. In response, we tested CIDD on additional targets: 1sa1, 1t8i and 7rpz, all known for their structural complexity and variability in ligand scaffolds. For 1sa1, Tubulin presents a challenging drug target because its binding sites are highly dynamic, involving extensive protein–protein interfaces; furthermore, the ligand in this complex, podophyllotoxin, is a complex natural product scaffold[1]. For 1t8i, Human DNA Topoisomerase I presents a challenging drug target because its catalytic mechanism involves a highly dynamic binding site that accommodates DNA; moreover, the ligand in this initial complex is camptothecin, a natural product scaffold rather than a traditionally rationally designed small molecule[2]. For 7rpz, KRASG12D is difficult to inhibit due to its high GDP/GTP affinity and absence of a nucleophilic residue near the switch II pocket, which hampers covalent strategies and limits prior chemotypes. The ligand uniquely overcomes these barriers with a pyrido[4,3-d]pyrimidine scaffold, fully engages the pocket, and achieves picomolar noncovalent inhibition with strong in vivo efficacy[3].
>
>
> Results on 1sa1:
>
> |  | Vina Score ↓ | QED ↑  | SA ↑   | MRR ↑   |  Success Ratio ↑ |
> |-----------|--------|------|------|---------|--------|
> |MolCRAFT| -9.78 | 0.56 | 0.66 | 35.56% |27.78% |
> |CIDD | -10.47 | 0.60 | 0.71 | 62.22%  | 54.44% |
>
> Results on 1t8i:
>
> |   | Vina Score ↓ | QED ↑  | SA ↑   | MRR ↑   |  Success Ratio ↑ |
> |-----------|-------|------|------|---------|-------|
> | MolCRAFT| -7.96| 0.37 | 0.57 | 60.24%  | 9.64%|
> | CIDD| -8.38 | 0.39 | 0.62 | 68.67%| 27.71%|
>
> Results on 7rpz:
>
> | | Vina Score ↓ | QED ↑  | SA ↑   | MRR ↑   |  Success Ratio ↑ |
> |--------|------------|------|------|---------|-----------|
> | MolCRAFT | -9.71 | 0.27 | 0.57 | 1.35% | 1.35% |
> | CIDD | -10.35 | 0.32 | 0.61 | 14.86%  | 14.86% |
>
>
> As shown, CIDD achieves substantially higher success ratios than the MolCRAFT baseline on these targets, even when the initial molecules generated by the basele model are of low quality. This demonstrates CIDD’s ability to be effective for chemically complex and challenging protein targets.
>
> We thank the reviewer again for this insightful suggestion, and we will include these new results and discussions in the revised paper.
>
> [1] Ravelli, R.B.G. et al. Nature 2004, 428, 198–202.
>
> [2] Staker, B.L. et al. J. Med. Chem. 2005, 48, 2336–2345.
>
> [3] Wang, X., et al. J. Med. Chem. 65.4 (2021): 3123–3133.
>
> ## Regarding prompt sensitivity
>
> We thank the reviewer for highlighting the importance of prompt sensitivity.
>
> First, as shown in Table 2(a), even when using different large language models (GPT-4o, GPT-4o-mini, DeepSeek-V3), the outcome metrics remain fairly consistent. This suggests that our framework is robust and not overly sensitive to the choice of underlying LLM. In other words, model updates or substitutions are unlikely to cause failures or unpredictable behaviors in the system.
>
> ### Table 2(a):Different LLM Backends in CIDD
>
> | LLM | Vina ↓  | MRR ↑    | Similarity ↑ |
> |---------------|---------|----------|----------------|
> | -   | -7.78   | 58.47%   | -  |
> | GPT-4o-mini| -8.29   | 80.02%   | 0.220 |
> | GPT-4o  | -8.50   | 81.37%   | 0.296 |
> | DeepSeek-v3 | -8.49   | 76.00%   | 0.379|
>
> To further assess prompt sensitivity, we conducted an additional experiment in which we asked ChatGPT to automatically rephrase the core design prompts, while preserving their original intent.  We use deepseek-v3 as our LLMs.
>
> As shown in the following table, the performance remained largely stable across the two prompt versions, indicating that the method is not highly sensitive to moderate rephrasings in Chain-of-Thought design instructions.
>
> |Case| Vina Score ↓ | QED ↑  | SA ↑   | MRR ↑   |  Success Ratio ↑ |
> |-----|---|------|------|------|-----|
> |ori prompt| -8.49| 0.565 | 0.714|76.0%|29.66%|
> |new prompt| -8.43| 0.581 | 0.728 |78.5%|29.67%|
>
> These results suggest that while prompt formulation can influence LLM responses, **CIDD remains robust under reasonable variations in prompt phrasing.**
>
> ## Regarding synthetic feasibility of our top candidates
>
> As shown in Table 1, the SA score reflects the synthetic feasibility of the generated molecules, and our method improves this metric compared to baselines.

---

> > ### Comment · Reviewer_M6aF · 2025-08-05
> >
> > Looks interesting! I like the additional experiment. Willing to increase the points.

---

> > > ### Author Response · Authors · 2025-08-07
> > > **Thank You for the Review**
> > >
> > > Dear Reviewer,
> > >
> > > We sincerely thank you for your thoughtful review, your positive feedback on our paper, and your engagement in the rebuttal process. Your feedback has certainly helped us improve our paper.

---

### Official Review · Reviewer_zFP8 · 2025-07-01

**Clarity:** 3
**Significance:** 3
**Originality:** 2
**Rating:** 5
**Confidence:** 4

**Summary:**

This work presents the Collaborative Intelligence Drug Design (CIDD) framework for LLM improvements to of small molecules generated by 3D structure-based drug design models. CIDD employs Chain-of-Thought (CoT) reasoning to iteratively analyze interaction profiles, propose substructural modifications, and select promising drug candidates. Additionally, the work introduces a new metric, the Molecule Reasonable Ratio (MRR) to illuminate the problem of improper aromatic ring hybridization states common to many 3D molecular generation models. The authors demonstrate that CIDD, when applied on top of pocket-based molecular generation models, achieves superior performance with respect to docking scores, traditional drug-likeness metrics like SA and QED score, and MRR compared to all baselines.

**Questions:**

1. Similarity is listed as a metric, but is it defined as the similarity of CIDD-generated candidates to the initial molecule or to each other?
2. Has CIDD been evaluated on hit-to-lead tasks? That is, given a reasonable binder posed in the protein pocket, can CIDD still generate superior molecules with respect to both potency and synthesizability?
3. Given an initial SBIG molecule, what is the time required for a single LEDD candidate to be generated by the LLM? Would it be viable to generate a screening set of, say, 1,000 or 10,000 molecules via LEDD for binding assays or more rigorous computational binding affinity calculations?
4. Are SA scores re-scaled for the results? The interpretation of the original SA score (between 1 and 10) was that higher scores meant more difficulty.

**Ethical Concerns:**

["NO or VERY MINOR ethics concerns only"]

**Final Justification:**

The approach is a simple yet effective way to obtain interpretable improvements to molecules generated by 3D generation methods. The authors adequately addressed the concerns and provided evidence that the model was capable of generating improvements even for ground-truth ligands.

**Limitations:**

Yes

**Quality:**

3

**Strengths And Weaknesses:**

Strengths:
1. The interaction and design steps encourage the LLM to make decisions grounded in chemical intuition by making explicit the various interactions between an initial docked ligand and the protein. The authors additionally demonstrate that interaction profiling is necessary comparing to PDB-formatted receptor data.
2. By design, CIDD is compatible with most 3-dimensional molecular generators that produce both a molecule and its pose in a target receptor.
3. The explicit rationale outlined in LEDD allows for clear interpretability in molecular design and optimization.
4. CIDD capitalizes on LLM tendencies to generate valid molecules to greatly improve MRR and other synthesizability metrics compared to initial designs.

Weaknesses:
1. In the experiments described, only 10 molecules are generated per protein pocket, making it difficult to draw conclusions about molecular diversity between LLM backends.
2. Interaction profiles calculated by the interaction analysis module rely on re-docking SBIG-generated molecules to the pocket, which can sometimes generate badly posed molecules that constitute poor starting points for optimization with LEDD.
3. Despite CIDD being a 2D molecular generation framework, the experiments omit comparisons against 2D generation methods guided by docking scores such as Cretu et al. (2024), Koziarski et al. (2024), Seo et al. (2024), Loeffler et al. (2024), and Swanson et al. (2025).


References

1. Cretu, M., Harris, C., Igashov, I., Schneuing, A., Segler, M., Correia, B., … Liò, P. (2025). SynFlowNet: Design of Diverse and Novel Molecules with Synthesis Constraints. arXiv [Cs.LG]. Retrieved from http://arxiv.org/abs/2405.01155
2. Koziarski, M., Rekesh, A., Shevchuk, D., van der Sloot, A., Gaiński, P., Bengio, Y., … Batey, R. A. (2024). RGFN: Synthesizable Molecular Generation Using GFlowNets. arXiv [Physics.Chem-Ph]. Retrieved from http://arxiv.org/abs/2406.08506
3. Loeffler, H.H., He, J., Tibo, A. et al. Reinvent 4: Modern AI–driven generative molecule design. J Cheminform 16, 20 (2024). https://doi.org/10.1186/s13321-024-00812-5
4. Seo, S., Kim, M., Shen, T., Ester, M., Park, J., Ahn, S., & Kim, W. Y. (2025). Generative Flows on Synthetic Pathway for Drug Design. arXiv [q-Bio.BM]. Retrieved from http://arxiv.org/abs/2410.04542
5. Swanson, K., Liu, G., Catacutan, D. B., McLellan, S., Arnold, A., Tu, M. M., … Stokes, J. M. (2025). SyntheMol-RL: a flexible reinforcement learning framework for designing novel and synthesizable antibiotics. bioRxiv. doi:10.1101/2025.05.17.654017

---

> ### Author Rebuttal · Authors · 2025-07-30
>
> We sincerely thank the reviewer for their thoughtful evaluation and positive feedback on our paper. We appreciate the insightful comments and are happy to address the points raised below.
>
> ## Regarding the diversity
>
> Thank you for pointing out the diversity metric. Initially, we included diversity in the main results table, but later decided to remove it as it did not offer significant insights—most methods showed only marginal differences in this aspect. For completeness, we provide the diversity scores of the baseline models and CIDD below:
>
> | Method       | LiGAN | AR   | Pocket2Mol | TargetDiff | IPDiff | DecompDiff | TAGMol | DrugGPS | MolCraft | CIDD | CIDD-100 |
> |--------------|-------|------|------------|-------------|--------|------------|--------|---------|----------|------|-----------|
> | Diversity    | 0.802 | 0.836| 0.866      | 0.890       | 0.887  | 0.877      | 0.893  | 0.908   | 0.870    | 0.867| 0.871     |
>
>
> We also evaluated the diversity when generating 100 molecules per target. As shown, CIDD achieves diversity comparable to its baseline model, MolCraft. Moreover, increasing the number of generated molecules from 10 to 100 does not significantly affect the average diversity, and diversity is not a concern of CIDD framework.
>
> ## Regarding re-docking SBIG-generated molecules
>
> We appreciate the reviewer’s insightful comment on the use of re-docking in our pipeline. The reason we perform re-docking, rather than directly using the generated molecular conformations, is that the original conformations produced by generative models may contain issues—such as atomic clashes or physically unstable geometries—that can negatively impact both interaction analysis and subsequent steps. This issue has been discussed in recent paper[1]. Re-docking helps ensure that we begin from a more stable and physically reasonable pose.
>
> That said, we fully agree with the reviewer that it is important to test whether this re-docking step is necessary. To this end, we conducted an ablation study where we directly used the originally generated conformations without re-docking. We use deepseek-v3 as our LLMs. The results are shown below:
>
> | Method     | Vina Score ↓ | QED ↑  | SA ↑   | MRR ↑   |  Success Ratio ↑ |
> |------------|--------------|--------|--------|---------|------------|
> | re-docking | -8.49      | 0.565 | 0.714 | 76.0%   | 29.66%         |
> | ori conformation | -8.403       | 0.575  | 0.721  | 76.09%  | 28.03%     |
>
> As shown, the performance differences are relatively small, suggesting that our framework is robust to this design choice. We thank the reviewer again for the valuable suggestion and will include this analysis and discussion in the appendix of the revised manuscript.
>
> ## Regarding 2D generation methods guided by docking scores
>
> Thank you for pointing us to these works. We appreciate the references and agree that those 2D molecular generation is an important and active line of research.
>
> We would like to clarify that our method is fundamentally different in formulation from the works cited. Specifically, CIDD is not an optimization framework: we do not use reinforcement learning, docking-score-based fine-tuning, or iterative optimization guided by oracle feedback. Instead, our framework focuses on repairing and improving imperfect 3D SBDD-generated molecules by preserving key protein-ligand interactions while enhancing drug-likeness and synthesizability—without relying on explicit reward signals or gradient guidance.
>
> Moreover, the referenced methods are not evaluated on the CrossDocked dataset, which we adopt as our main benchmark to ensure consistency with our chosen baselines. Likewise, these methods are not included as baselines in the prior works we compare against. As a result, a fair and meaningful comparison would require non-trivial adaptation and retraining of those methods in our setting—especially considering that some of them are specifically designed for narrow domains such as antibiotic discovery—which is unfortunately infeasible within the limited rebuttal period. In addition, several of the referenced methods can be computationally intensive, which is why they are typically evaluated only on a small number of selected targets.
>
> Importantly, we view CIDD as complementary to these optimization-based methods. In fact, many of them could be incorporated into our pipeline as components within the SBIG (Structure-Based Initial Generation) stage. Our framework is modular by design, and can serve as a downstream interaction-preserving refinement module, enhancing the outputs of other generative methods that may not explicitly account for protein context or structural plausibility.
>
> We thank the reviewer again for bringing these methods to our attention. We will cite the relevant works and include a discussion of their methodological differences, as well as their relationship to CIDD, in the Related Work section of the revised manuscript.
>
> ## Regarding similarity metrics
>
> Thank you for pointing out the metric on similarity. Yes, we do evaluate the similarity between the raw molecule from the SBIG module and the final molecule generated by CIDD. This choice is intentional and aligned with the design goal of the LEDD step: to make minimal but meaningful modifications that improve chemical quality while preserving the structural basis provided by the SBDD model.
>
> Specifically, we treat the SBIG-generated molecule as a structure-based proposal that carries important protein-ligand interaction information. Rather than generating a completely new molecule, which may disrupt these interactions, we aim to refine the initial structure in a way that retains its interaction intent. Therefore, a higher similarity reflects a successful case where the molecule is improved in terms of drug-likeness or synthesizability, while maintaining the original design.
>
> This design choice also supports a data-centric view. By producing pairs of molecules with small structural differences but large differences in quality, our framework naturally generates valuable data for future training of optimization models. These minimally perturbed, property-improved pairs can serve as useful examples for learning structure-to-property relationships.
>
> We will clarify the motivation and implications of this similarity metric in the revised version of the manuscript.
>
> ## Regarding hit-to-lead tasks
>
> We thank the reviewer for highlighting the potential of applying the CIDD framework to hit-to-lead optimization. Although CIDD was not originally designed for this purpose, it can indeed be adapted to such scenarios. To explore this, we initialized the generation process—and the input to the LEDD module—using ground-truth ligands from the CrossDocked test set. As shown in the following table, despite these ligands already exhibiting good synthetic accessibility and a high reasonable ratio, CIDD was still able to generate molecules with further improvements in potency, synthesizability, and structural reasonability, demonstrating its effectiveness and potential in hit-to-lead optimization.
>
> |          | Vina Score ↓ | QED ↑  | SA ↑   | MRR ↑   |  Success Ratio ↑ |
> |-----------|------------|------|------|---------|----------------|
> | Ori ligands  | -7.53      | 0.473 | 0.738 | 0.8272  | 22.22%         |
> | CIDD gen| -8.22      | 0.528 | 0.752 | 0.9136  | 37.04%         |
>
> Notably, as we mentioned, the original CIDD (LEDD) framework was not specifically designed for hit-to-lead optimization tasks. Fully supporting such applications would require adjustments to the pipeline design, modifications to the chain-of-thought prompts, and changes in tool integration. Nonetheless, the current results already demonstrate **the potential of our proposed domain model and LLM collaboration paradigm in broader scenarios such as hit-to-lead**. We will incorporate this direction into the discussion of future work.
>
> We sincerely thank the reviewer again for the valuable feedback.
>
> ## Regarding generating 1,000 or 10,000 molecules
>
> Thank you for the thoughtful question. We confirm that the CIDD pipeline supports large-scale candidate generation, including sets of 1,000 or even 10,000 molecules.
>
> The end-to-end generation time for a single molecule—including interaction parsing, prompt construction, LLM inference, and generating for N=5 new designs—is approximately 5 minutes when run sequentially. However, the pipeline is **inherently parallelizable**. When deployed via cloud-based API services (such as OpenAI or DeepSeek), there is virtually no limit on parallelization, assuming adequate API throughput. In practice, this allows for generating 1,000 molecules **within just a few minutes** using parallel asynchronous requests.
>
> Compared to the time required for downstream experiments such as molecular dynamics simulations, or wet-lab synthesis and validation, this generation time remains extremely efficient, and does not constitute a computational bottleneck in the overall drug discovery pipeline.
>
> ## Regarding SA scores
>
> Thank you for the question. Yes, we re-scaled the original synthetic accessibility (SA) scores for consistency and interpretability in our evaluation.
>
> The original SA score ranges from 1 (easy to synthesize) to 10 (difficult to synthesize). To make the interpretation consistent with other normalized scores (where higher is better), we applied the following linear transformation:
>
>
>     sa_norm = round((10 - sa) / 9, 2)
>
>
> This maps the SA score to a normalized range between 0 and 1, where a higher value indicates easier synthesis. This rescaled score was used for reporting purposes in our tables and analysis.

---

> ### Comment · Reviewer_zFP8 · 2025-08-05
>
> This reviewer would like to thank the authors for their detailed response.
>
> The authors highlight that the method makes interaction-preserving repairs and improvements to molecules generated by the SBIG. While it is true that the proposed method does not rely on reward signals directly, the interactions are hypothesized via either re-docking or the original generated structure, both of which are dubious assumptions.
>
> Despite this, it is compelling that the method is able to obtain cheap and substantial improvements to molecules without additional training, and this reviewer finds the paired-data idea for training future structure-to-property models an interesting application. For this reason, the score will be raised to 5.

---

> > ### Author Response · Authors · 2025-08-07
> > **Thank You for the Review**
> >
> > Dear Reviewer,
> >
> > We sincerely thank you for your thoughtful review, your positive feedback on our paper, and your engagement in the rebuttal process. Your feedback has certainly helped us improve our paper.

---

### Official Review · Reviewer_hXF6 · 2025-07-02

**Clarity:** 4
**Significance:** 4
**Originality:** 4
**Rating:** 5
**Confidence:** 4

**Summary:**

This paper accurately identifies the drawback of current 3D generative models: they often produce unreasonable molecular structures. Firstly, this paper proposes MRR as a novel metric, which evaluates molecules from diverse dimensions. Then, considering the great potential of LLMs in designing reasonable molecules and excellent binding affinity provided by 3D generative models, the authors propose CIDD to combine these complementary worlds. To be specific, a two-stage pipeline is established to generate initial candidates and further optimize them. Extensive experiments on CrossDocked2020 confirm the effectiveness of the CIDD pipeline, showing great potential of this work in drug design.

**Questions:**

1. Why is 3DSBDD not compared in Tab. 1? This should be a meaningful baseline.
2. Following Q1, does the Vina score of generated molecules drop after modifications from LLMs?
3. You claim to apply MolCRAFT in the SBIG step, while in Tab.1, the method is named as "3DSBDD + LLM". Why is that?

**Ethical Concerns:**

["NO or VERY MINOR ethics concerns only"]

**Final Justification:**

I believe this is a good work, and maintain my score.

**Limitations:**

Yes.

**Quality:**

4

**Strengths And Weaknesses:**

Strengths:
1. This paper aims to solve the most challenging dilemma in SBDD: the lack of reasonable structures. More surprisingly, this paper combines LLMs as a knowledge base to optimize generated molecules.
2. Experiments are rigorously executed. Ablation studies are convincing, showing CIDD can be generalized to multiple SBDD methods and LLMs.
3. Experiments show promising results. 3DSBDD+LLM shows overall SOTA performance on six metrics. Moreover, it generates plausible molecules with larger molecular weights, achieving the best balance between binding affinity and reasonable structures.
4. MRR seems to be rational, considering multiple useful chemical aspects (although I am not an expert in this field). It can serve as a better metric than QED, SA, etc., boosting the entire community.

Weaknesses:
1. CIDD does not directly generate molecules with 3D conformations, which may take a little bit longer time to perform docking and other conformational tasks.
2. No code is provided.

---

> ### Author Rebuttal · Authors · 2025-07-30
>
> We sincerely thank the reviewer for their thoughtful evaluation and positive feedback on our paper. We appreciate the insightful comments and are happy to address the points raised below.
>
> ## Regarding CIDD not directly generating 3D conformations and taking longer computation time
>
> We acknowledge the reviewer’s observation that our proposed CIDD framework may incur longer computational time compared to traditional 3D structure-based drug design (SBDD) methods. While this is indeed a relevant concern, we would like to offer additional context to clarify this trade-off.
>
> Drug discovery is inherently a **time-intensive and resource-demanding process**, often spanning several years and incurring significant financial costs. A substantial portion of this effort is driven by human tasks—such as expert assessments, manual filtering, and iterative decision-making—rather than computational bottlenecks alone.
>
> CIDD explores a different trade-off: although it introduces additional computational steps, it significantly **reduces human involvement** by leveraging large language models (LLMs) to enhance chemical reasonability and drug-likeness. These are aspects that traditionally rely on deep domain expertise. In our experiments, CIDD consistently **improves the quality of generated molecules** in these dimensions, delivering results that are more chemically reasonable with less need for manual inspection.
>
> We believe that, in practice, this translates to a net gain in efficiency: **the additional generation time is more than offset by the reduction in expert labor and manual validation efforts.**
>
> We appreciate the reviewer’s observation and will incorporate this discussion into the limitations section of our revised manuscript.
>
> ## Regarding the code
>
> We apologize for not providing the codebase during the initial submission. We are fully committed to releasing the code and supporting reproducibility. However, due to file-upload restrictions during the rebuttal phase, we are currently unable to share the code publicly. We will release the full codebase upon acceptance.
>
> That said, in LLM-based systems like ours, **prompt design** plays a central role in governing the model’s reasoning behavior. In the CIDD framework, prompts dictate how the LLM interprets structural context, coordinates tool usage, and executes multi-step molecular design decisions. As such, we have placed particular emphasis on reproducibility through prompt transparency.
>
> To support this, **we provide detailed prompt templates and example model responses for each stage of the pipeline in the Appendix.** Coupled with the step-by-step descriptions in the main text, we believe these materials offer sufficient detail for readers to reproduce and understand the behavior of our system, even prior to code release.
>
> If there are any ambiguities or questions regarding the prompts we’ve provided, we are more than happy to clarify them during the rebuttal process.
>
> ## Regarding the 3D-SBDD baselines
>
> We apologize for any confusion caused by the terminology and formatting used in the table. We understand the reviewer’s concern that “3DSBDD + LLM” may be interpreted as an individual method rather than a category.
>
> To clarify: in our work, “3DSBDD” refers to the collection of **all baseline methods** we compare against, rather than a specific standalone approach. In particular, we do employ **MolCRAFT as our SBIG (structure-based Interation Generation) module** to generate the initial raw molecules, with the corresponding results shown in Table 1. And as shown in the table, the Vina score of generated molecules **actually becomes better compared to MolCRAFT(the second-to-last line in Tab. 1)** after modifications from LLMs. （-8.496 vs -7.783）
>
> To avoid ambiguity, we will revise the table caption and terminology to make this categorization clearer and more precise in the updated version.
>
> Thank you again for pointing this out—your careful review and valuable feedback have been very helpful in improving the clarity and rigor of our work.

---

> > ### Comment · Reviewer_hXF6 · 2025-08-06
> > **Response**
> >
> > Thanks for further clarification. I decide to maintain my score. Good luck!

---

> > > ### Author Response · Authors · 2025-08-07
> > > **Thank You for the Review and a Kind Reminder About Final Justification**
> > >
> > > Dear Reviewer,
> > >
> > > We sincerely thank you for your thoughtful review, your positive feedback on our paper, and your engagement in the rebuttal process.
> > >
> > > As a gentle reminder, even if your score remains unchanged, the system requires a brief final justification. We would greatly appreciate it if you could update your review by clicking the Edit button and completing the “Final Justification” section.
> > >
> > > We deeply appreciate your time and support.

---

### Decision · Program_Chairs · 2025-09-17

**Decision:**

Accept (poster)

**Comment:**

This work presents the CIDD framework for small-molecule generation. The experiments are rigorous, the metrics are meaningful and the ablation studies are convincing.  Reviewers generally agreed that the problem framing is relevant and that the proposed metric is clearly defined. Remaining concerns focus mainly on validation and external oracles. The rebuttal added useful analyses and clarifications that addressed several points, though questions about efficiency and experimental validation are only partially resolved. I find this to be an inspiring multimodal study that combines SBDD with LLM reasoning, and overall I tend to support its acceptance.